# Temperature and VOC concentration as controlling factors for chemical composition of alpha-pinene derived secondary organic aerosol

Louise. N. Jensen[1], Manjula. R. Canagaratna[2], Kasper Kristensen[3], Lauriane L. J. Quéléver[4], Berndette

Rosati[1,5], Ricky Teiwes[5], Marianne Glasius[1], Henrik B. Pedersen[5], Mikael Ehn[4] and Merete Bilde[1]

[1]Department of Chemistry, Aarhus University, 8000 Aarhus C, Denmark
[2]Aerodyne Research, Inc., Billerica, MA, USA
[3]Department of Engineering, Aarhus University, 8000 Aarhus C, Denmark
[4]Institute for Atmospheric and Earth System Research – INAR / Physics, P.O. Box 64, FI-00014, University of Helsinki,
Finland
[5]Department of Physics and Astronomy, Aarhus University, 8000 Aarhus C, Denmark

*Correspondence to*: Merete Bilde (bilde@chem.au.dk)

**Abstract**

This work investigates the individual and combined effects of temperature and volatile organic compound precursor
concentration on the chemical composition of particles formed in the dark ozonolysis of α-pinene. All experiments were
conducted in a 5 m$^3$ Teflon chamber at an initial ozone concentration of 100 ppb and initial α-pinene concentrations of 10 ppb
and 50 ppb, respectively, at constant temperatures of 20 °C, 0 °C, or -15 °C, and at changing temperatures (ramps) from
-15 °C to 20 °C and from 20 °C to -15 °C. The chemical composition of the particles was probed using a High-Resolution
Time-of-Flight Aerosol Mass Spectrometer (HR-ToF-AMS).

A four-factor solution of a Positive Matrix Factorization (PMF) analysis of the combined HR-ToF-AMS data is presented. The
PMF analysis as well as elemental composition analysis of individual experiments show that secondary organic aerosol
particles with the highest oxidation level are formed from the lowest initial α-pinene concentration (10 ppb) and at the highest
temperature (20 °C). Higher initial α-pinene concentration (50 ppb) and/or lower temperature (0 °C or -15 °C) result in lower
oxidation level of the molecules contained in the particles. With respect to carbon oxidation state, particles formed at 0 °C are
25 more comparable to particles formed at -15 °C than to those formed at 20 °C. A remarkable observation is that changes in
temperature during particle formation result in only minor changes in the elemental composition of the particles. The
temperature at which aerosol particle formation is induced, thus seems to be a critical parameter for the particle elemental
composition.

Comparison of the HR-ToF-AMS derived estimates of the content of organic acids in the particles based on m/z 44 in the mass
spectra show good agreement with results from off-line molecular analysis of particle filter samples collected from the same
experiments. Higher temperatures are associated with a decrease in the absolute mass concentrations of organic acids (R-
COOH) and organic acid functionalities (-COOH), while the organic acid functionalities account for an increasing fraction of
the measured particle mass.

**1 Introduction**

Atmospheric aerosol particles can alter air quality (WHO 2016) and visibility (Wang et al. 2009) on a regional scale. On a
global scale, particles affect cloud formation, the radiative balance, and thus climate (IPCC 2013).

Atmospheric particles are chemically diverse entities, often with a significant mass fraction of organic compounds (Zhang et
al. 2007; Jimenez et al. 2009). Secondary Organic Aerosol (SOA) is formed from condensation of oxidation products of volatile

organic compounds (VOC) emitted from both anthropogenic and biogenic sources (Sindelarova et al. 2014; Seinfeld and Pandis 2016). α-pinene is a biogenic VOC emitted from e.g. foliage of coniferous trees (Rasmussen 1972), and it has been identified as the most common monoterpene in boreal forests all year round (Hakola et al. 2003). In the atmosphere, α-pinene is oxidized primarily by ozone ($O_3$), hydroxyl radicals (OH·), and nitrate radicals ($NO_3$·). Due to their low vapor pressures, some of the gas phase oxidation products may partition onto already existing particles by condensation or reactive uptake, and contribute to particle growth (Hallquist et al. 2009). In addition, some low vapor pressure oxidation products of α-pinene are able to nucleate (Kirkby et al. 2016) and likely play an important role in the initial growth of new particles in the atmosphere (O'Dowd et al. 2002; Riipinen et al. 2012; Ehn et al. 2014; Tröstl et al. 2016).

It is well established that the particle mass available for condensation of gases affects the partitioning of organic species between gas phase and particle phase (Pankow 1994a, 1994b) although, the traditional partitioning theory is limited in relation to non-liquid, more viscous particles, such as α-pinene derived SOA formed at low relative humidity (Renbaum-Wolff et al. 2013), because of slow diffusion (Cappa and Wilson, 2011; Pöschl, 2011).

The fraction (F) of a given semi-volatile species in the particle phase at a given temperature has been formulated in an absorptive equilibrium partitioning framework as

$$F = \frac{1}{1 + c^*/M} \qquad (1)$$

where c* is the gas phase mass concentration at saturation and M is the mass concentration of absorbing material (Kroll and Seinfeld 2008). Thus, the chemical composition of a particle that is in equilibrium with the surrounding gas phase is affected by both c* and M. The c* of a gaseous compound is generally inversely related to its level of oxidation (Jimenez et al. 2009). The particle composition can be shifted towards species with higher c* values (i.e. less oxidized, more volatile species) by increasing the mass concentration of pre-existing particles, i.e. the value of M; conversely, lower M values result in particle phase compositions that are dominated by species with lower c* values (i.e. more oxidized, less volatile species). This has been experimentally confirmed by e.g. Shilling et al., (2009), who showed that the oxidation level of SOA from α-pinene ozonolysis decreases with increasing particle mass loadings.

The equilibrium partitioning of a gas with a given c* (volatility) also depends on temperature, as demonstrated by Pathak et al. (2007), Saathoff et al. (2009), and Warren et al. (2009), based on chamber studies of α-pinene derived particles at different constant temperatures between -30 °C and 45 °C. Partitioning has also been addressed in chamber studies where the temperature was ramped after the initial (constant temperature) formation of SOA. Stanier et al. (2007) increased the temperature from 22 °C to a maximum of 40 °C, and in some experiments decreased the temperature back to 22 °C. During heating, they observed a decrease in SOA size, an indication of evaporation, and during cooling an increase in SOA size, an indication of condensation. In experiments by Warren et al. (2009), where the temperature was cycled in the ranges of 5 °C to 27 °C and 27 °C to 45 °C, heating was associated with a decrease in particle mass, and cooling associated with an increase in particle mass. In a recent study by Zhao et al. (2019), where the temperature was cycled between 5 °C and 35 °C (RH varied between 10 and 80 %), it is suggested, that condensation during cooling is smaller than predicted by equilibrium partitioning.

The chemical composition of gas and particle phase in α-pinene ozonolysis experiments is determined by a combination of thermodynamic and kinetic aspects (Zhang et al. 2015; Kristensen et al. 2017). The effect of temperatures below room temperature (~20 °C), in particular below 0 °C, on gas phase oxidation products, nucleation, SOA growth, and particle chemical composition, however, remains a largely unexplored area (Kristensen et al. 2017, Simon et al. 2020). Since low temperatures are of high atmospheric relevance, e.g. low temperatures are prevailing at the latitudes of the boreal forests and at higher elevation, it is important to quantify SOA formation and properties under cold conditions. Furthermore, vertical transport can lead to changes in temperature within short time frames, affecting reaction kinetics, condensation processes, and properties relevant for the climate effect of particles (Topping et al. 2013; Murphy et al. 2015).

The lack of knowledge on how the chemical composition of both the gas phase and particle phase vary with temperatures was the motivation behind the Aarhus Chamber Campaign on HOMs and Aerosols (ACCHA) introduced in Kristensen et al. (2020). The ACCHA campaign focuses on temperatures from 20° C to -15° C, corresponding to conditions relevant in the boreal forest regions (Portillo-Estrada et al. 2013). As in most chamber experiments VOC concentrations in the ACCHA campaign (10-50 ppb) were one to two orders of magnitude higher than typical ambient conditions (Kourtchev et al. 2016). These conditions were chosen to speed up aerosol formation in the experiments and we believe the data provides valuable and atmospherically relevant information, applicable to e.g. the boreal forest areas.

The impact of temperature on the yield of highly oxygenated organic molecules (HOMs) is presented in Quéléver et al. (2019), and more details on the volatile organic compounds are presented in Rosati et al. (2019).

The goal of the current paper is to investigate and quantify the individual and combined effects of α-pinene precursor concentration and temperature on SOA mass concentration and chemical composition. For this purpose, we here describe and discuss a subset of the data collected during the ACCHA campaign, focusing on results obtained from a High-Resolution Time-of-Flight Aerosol Mass Spectrometer (HR-ToF-AMS).

## 2 Methods

### 2.1 Experimental

This work is based on experiments conducted in the Aarhus University Research on Aerosol (AURA) chamber; a ~5 m³ bag made of 125 µm FEP Teflon film located in an enclosure, where the temperature is controllable between -16 °C and 26 °C. The AURA chamber has been described in detail by Kristensen et al. (2017).

The experiments were conducted as part of the ACCHA campaign and focus on SOA formed in dark ozonolysis of α-pinene at various temperatures. An overview of the campaign is provided in Kristensen et al. (2020). To ease the reading of the current paper, a short summary of the ACCHA campaign is given here, and a modified version of the overview table of the experiments from Kristensen et al. (2020), is presented in Table 1 where focus is on the parameters relevant in this work,. At a constant temperature of either 20 °C, 0 °C, or -15 °C, ozone was injected into the chamber to a concentration of ~100 ppb, followed by injection of either 10 ppb (low concentration) or 50 ppb (high concentration) α-pinene. The chamber was operated at atmospheric pressure, and neither seed particles nor OH-scavengers were introduced.

Three series of constant temperature experiments, all consisting of an experiment at 20 °C, 0 °C, and -15 °C, were conducted. In one of the series 10 ppb α-pinene was injected into the chamber (experiments 1.1-1.3), while two similar series of experiments were performed at 50 ppb α-pinene (experiments 2.1-2.3 and 3.1-3.3). Additionally, the series of 10 ppb α-pinene experiments includes two temperature ramp experiments, where the temperature was decreased from 20 °C to -15 °C (experiment 1.4) and increased from -15 °C to 20 °C (experiment 1.5) ~35 minutes after α-pinene injection, which corresponds to the period during SOA formation and before mass peak.

In this work, we present data from a subset of instruments involved in the ACCHA campaign: a temperature and humidity sensor (HC02-04) attached to a HygroFlex HF320 transmitter (Rotronic AG) placed in the center of the chamber, a scanning mobility particle sizer (SMPS), consisting of a Differential Mobility Analyzer (DMA, TSI 3082) and a nano water-based condensation particle counter (CPC, TSI 3788), and a High-Resolution Time-of-Flight Aerosol Mass Spectrometer (HR-ToF-AMS, Aerodyne Research Inc.) (Jayne et al. 2000; DeCarlo et al. 2006; Canagaratna et al. 2007). In the following, the HR-ToF-AMS will be referred to as AMS. Both the SMPS and AMS were placed at room temperature next to the chamber outlets, and the connecting tubing was temperature insulated.

By the end of each experiment, a particle sample was collected on a Teflon filter (0.45 µm pore size, Chromafil). Particle samples were extracted and analyzed by an Ultra High Performance Liquid Chromatograph/Electrospray Ionization quadrupole Time-of-Flight Mass Spectrometer (UHPLC/ESI-qTOF-MS, Bruker Daltonic), as described in Kristensen et al. (2020), where also the analytical method and results are presented in detail. Herein, we compare the findings from the
5   UHPLC/ESI-qTOF-MS, hereafter referred to as LC-MS, to the AMS measurements.

## 2.2 Data analysis

Positive Matrix Factorization (PMF) (Paatero and Tapper 1994; Paatero 1997) has traditionally been used to investigate contributions of different sources to ambient particles, and the application of PMF to AMS data from chamber experiments was first demonstrated by Craven et al. (2012). In the present work, PMF analysis is applied to chemical composition data
10   from SOA particles that are produced in the ozonolysis of α-pinene, but formed and aged under different temperatures and precursor concentrations, and consequently different particle loadings. High-resolution AMS mass spectra of SOA particles from the various experimental conditions were analyzed in one matrix, allowing for monitoring spectral and elemental chemical composition changes that occur as conditions change. The PET tool (V 2.09A) was used to perform the PMF analysis on high-resolution AMS mass spectra, according to the principles described in detail by Ulbrich et al. (2009).

PMF is a model that can be used to express measured mass spectra as a linear combination of factors that are the products of constant mass spectra and related time profiles as follows:

$$x_{ij} = \sum_p g_{ip} f_{pj} + e_{ij} \qquad (2)$$

The measured mass spectral data is the matrix X, an $m \times n$ matrix with $n$ ion masses measured at $m$ different time points, and $x_{ij}$ is an element of this matrix. $p$ is the number of factors chosen for the solution, $g_{ip}$ is an element of the matrix G containing time series of the factors, and $f_{pj}$ is an element of the matrix F of constant factor mass spectral profiles. The matrix elements $e_{ij}$ correspond to the error matrix, E, of residuals not explained by the model (Paatero 1997; Ulbrich et al. 2009). Equation (2) is
solved using the PMF2 algorithm (Paatero 1997), which uses linear least-squares fitting together with the constraints that the values of matrix F and G have to be non-negative. The solution is found by minimizing the fit parameter Q:

$$Q = \sum_{i=1}^{m} \sum_{j=1}^{n} \left( \frac{e_{ij}}{\sigma_{ij}} \right)^2 \qquad (3)$$

where $\sigma_{ij}$ is an element of a matrix containing the standard deviations for each element of X (Paatero 1997; Ulbrich et al. 2009). Estimation of standard deviations was performed as outlined in Ulbrich et al. (2009) with "weak" ions (i.e. ions with signal/noise (S/N) < 2) being down weighted by a factor of two, and "bad" ions (i.e. ions with S/N < 0.2) being down weighted by a factor of ten. Additional sources of uncertainty that are not accounted for in the PMF analysis of high-resolution mass spectra, are uncertainties related to high-resolution fitting, including errors in peak shape, and m/z calibrations (Cubison et al.
(2015). The number of factors ($p$) is chosen based on a combination of evaluation of residuals, Q values, and a priori knowledge about the dataset (Lanz et al. 2007; Ulbrich et al. 2009). In the result section, a four-factor solution of the PMF analysis of high-resolution AMS data is presented. Although the five-factor solution and six-factor solution have lower Q/Q$_{expected}$ (Q$_{expected}$ $\approx m \times n$, i.e. the number of points in the data matrix (Ulbrich et al. 2009)) compared to the four-factor solution, the larger number of factors are not selected because they do not provide any more interpretable information about the particle

composition. The background for choosing the four-factor solution over the five-factor and six-factor solutions is explained more detailed in the supplementary information (S1-S11).

Since previous laboratory experiments show that the collection efficiency (CE) and relative ionization efficiency (RIE) of laboratory SOA are variable (Docherty et al. 2013), the mass concentrations presented in the PMF analyses are estimated from the total SOA mass concentration, as obtained from integrated SMPS size distributions, assuming spherical particles and densities calculated from the AMS derived elemental ratios (Kuwata et al. 2011). Densities are derived as averages based on AMS data from the last 30 minutes of each experiment. Uncertainties related to the density calculation are described in Kuwata et al. (2011).

Mass spectra of the factors from the PMF analysis are compared using the methods described by Wan et al. (2002) and Ulbrich et al. (2009), respectively. The comparisons focus on both the entire high-resolution mass spectra, m/z 12 to m/z 115, and the range of m/z > 44 to prevent an impact from the most intense peaks, especially m/z 43 and m/z 44, which are the ones associated with the largest variation between the factors.

Low SOA concentrations in the beginning of the experiments increase the uncertainty of the AMS measurements. Therefore, the first 4 to 16 minutes of the experiments (longest in the 10 ppb α-pinene experiments) are omitted from the elemental analysis of the AMS data.

## 3 Results and discussion

To provide an overview of the course of a typical experiment, Figure 1a shows the evolution in particle mass concentration and the elemental composition, illustrated by the oxygen-to-carbon (O:C) ratio in experiment 2.3 which is conducted at -15 °C with an initial α-pinene concentration of 50 ppb. In the beginning of the experiment, both the mass concentration and O:C ratio increase significantly, but after ~50 minutes, the O:C ratio tend to stabilize while the particle mass concentration continues increasing and peaks after ~175 minutes (not corrected for wall loss). The reported particle mass concentration is obtained using a density of 1.12 g cm$^{-3}$. According to Table 1 and supplementary material S12, by the end of the experiments the AMS derived SOA densities are of the order 1.1-1.3 g cm$^{-3}$. There are indications of a slight increase in density with higher experimental temperature as well as a slightly higher density for the particles formed at low α-pinene concentration (10 ppb) compared to high α-pinene concentration (50 ppb). For reproducibility with respect to SOA formation (mass concentration) as well as loss rates of α-pinene and ozone, see Kristensen et al. (2020).

Figure 1b is a mass spectrum of experiment 2.3 obtained from the high-resolution AMS data at the highest particle mass concentration (not wall loss corrected). It shows that fragments, which belong to the so-called hydrocarbon family (CH), are distributed throughout the mass spectrum, with some of the most prominent peaks (and ions) being m/z 39 ($C_3H_3^+$), 41 ($C_3H_5^+$), and 55 ($C_4H_7^+$). The oxidized compounds, which belong to the CHO1 and CHOgt1 (gt means greater than) families, dominate at m/z 28 (estimated from $CO_2^+$ according to Aiken et al. (2008)), 29 ($CHO^+$), 43 ($C_2H_3O^+$), 44 ($C_2H_4O^+$, $CO_2^+$), 55 ($C_3H_3O^+$), and 83 ($C_5H_7O^+$). m/z 43 is the most significant peak and it also has the highest contribution of the CHO1 family, while the more oxidized CHOgt1 family dominates in m/z 44. Previous studies have shown that these two peaks provide useful information about particle oxidation level (Ng et al. 2011). The patterns described above are also observed in the mass spectra of the other experiments, which are overall highly comparable across experimental conditions (supplementary material S13-S23) as well as comparable to mass spectra of particles formed in dark ozonolysis of α-pinene in other chambers (Bahreini et al. 2005; Song et al. 2007; Shilling et al. 2009; Chhabra et al. 2010).

### 3.1 PMF analysis of Aerosol Mass Spectra

While PMF analysis is traditionally utilized to identify distinct sources in ambient measurements and the factors are named according to what they are (e.g. Oxygenated Organic Aerosol, OOA), here PMF analysis of the combined dataset of α-pinene

SOA experiments provides a tool for identifying subtle changes in the measured mass spectra across the different experimental conditions and the factors are named according to the conditions under which they dominate. The analysis was performed on a combined dataset representing eight different experimental conditions: three constant temperature conditions, 20 °C, 0 °C, and -15 °C, with two initial α-pinene concentrations, 10 ppb and 50 ppb (experiments 1.1, 1.2, 1.3, 3.1, 3.2, and 3.3), and two temperature ramps, from 20 °C to -15 °C and from -15 °C to 20 °C, both with an initial α-pinene concentration of 10 ppb (experiments 1.4 and 1.5).

The result of the four-factor solution from the PMF analysis is presented in Figure 2 (and S7), showing the changes in mass concentration of the factors as a function of time in each experiment, and in Figure 3, showing the high-resolution mass spectra of the four factors, i.e. factor profiles. The mass spectra are colored according to contributions from the various types of elemental compositions (i.e. ion families) that appear at each ion signal. As expected from the comparison of the mass spectra from the individual experiments (supplementary material S13-S23), the factor profiles show a high degree of similarity (Figure 3 and S24) with small differences in the relative intensities of ions. Using the method described by Ulbrich et al. (2009), on a scale from 0 to 1, with 1 indicating highest similarity, the similarity is calculated to be between 0.86 and 0.97 across all factors in the m/z range from 12 to 115 with Factor 3 and Factor 4 being the most similar and Factor 1 and 3 being least similar. By focusing only on the m/z > 44, similarities between 0.85 and 0.97 are obtained and Factor 2 and Factor 3 are the most similar and Factor 1 and Factor 2 are least similar. S24 also show the corresponding results of a comparison using the method described by Wan et al. (2002).

Table 2 summarizes how the four factors do differ in oxygen-to-carbon (O:C) ratio, hydrogen-to-carbon (H:C) ratio, average carbon oxidation state ($OS_C$, Kroll et al. 2011) as well as the ratios between the absolute intensities of the fragment ions at m/z 43 ($C_3H_7^+$, $C_2H_3O^+$) and m/z 44 ($C_2H_4O^+$, $CO_2^+$), respectively, and the corresponding total organic ion intensity ($f_{43}$ and $f_{44}$, respectively). Variations in these parameters can help explain the relative contribution of each factor to the SOA mass under different experimental conditions (i.e. temperatures and α-pinene concentrations (particle mass loadings)).

As shown in Figure 2, the SOA observed under each experimental condition does not correspond to a single PMF factor. The SOA obtained under different experimental conditions are represented by a linear combination of multiple factors instead. The factors are clearly distinguished from each other, however, by consistent trends in their relative mass contributions to the SOA observed under the different experimental conditions. These trends are used in the interpretation and naming of the factors. According to their appearance and relative contribution to total SOA mass, Factors 1 and 4 will in the following discussion be referred to as "temperature factors", and Factors 2 and 3 will be referred to as "concentration factors". For example, according to Figure 2, at both α-pinene concentrations Factor 1 makes up a significant fraction of the particle mass in the 20 °C experiments, but plays a minor role in the colder experiments. Therefore, Factor 1 will be referred to as "high temperature factor". The significant contribution of Factor 1 to the SOA mass at high temperature is in agreement with the fact that this factor is mostly dominated by ions from oxidized species (i.e. high intensity of CHO1 and CHOgt1 ion groups at m/z 28, 29, 43, 44, 55, and 83) (Figure 3). Among all factors, Factor 1 has the highest O:C ratio (0.56), $OS_C$ (-0.53), $f_{43}$ (14 %), and $f_{44}$ (9 %) (Table 2), and therefore the chemical species represented by Factor 1 (i.e. related to high temperature) are likely the most oxidized entities present in the SOA.

Factor 2 is dominating in all 10 ppb α-pinene experiments, but is almost nonexistent in the 50 ppb α-pinene experiments. It will therefore be referred to as a "low concentration factor" (Figure 2). Among all factors, Factor 2 (low α-pinene concentration) has the second highest contribution of oxidized ions (CHO1 and CHOgt1 family) (Figure 3), as well as the second highest O:C ratio (0.39), $OS_C$ (-0.81), and $f_{44}$ (8 %) (Table 2). With $f_{43}$ (7 %) being of similar magnitudes as $f_{44}$, it has the highest relative ratio of $f_{44}$-to-$f_{43}$ among the four factors. Furthermore, Factor 2 also has strong contributions from hydrocarbons (CH family) such as m/z 39, 41, 55, 67, 69, 79, 81, and 91 (Figure 3).

Factor 3 makes up a significant fraction of the particle mass formed in the 50 ppb α-pinene experiments at all temperatures. It plays a less important role in the 10 ppb α-pinene experiments conducted at both 0 °C and -15 °C, whereas at 20 °C, where the

lowest particle mass is formed, Factor 3 is practically nonexistent (Figure 2). Based on this appearance, Factor 3 will be referred to as a "high concentration factor". Looking into the profile of Factor 3 (high α-pinene concentration), it has relatively high contributions from the CHO1 family at m/z 55 and m/z 83, and among all factors, it has the highest contribution of m/z 91 from the CH family, which has been used as a tracer of biogenic emissions in ambient measurements (Lee et al. 2016) (Figure

3). Factor 3 also has the lowest $f_{44}$ (3 %), O:C (0.26), and $OS_C$ (-1.08) among all factors, i.e., it represents the least oxidized material (Table 2).

Factor 4 (low temperature) appears at low temperature in SOA formed in both 10 ppb and 50 ppb α-pinene experiments (Figure 2). It has around the same level of $f_{43}$, (13 %) as Factor 1 and relatively high intensities of m/z 55 and m/z 83, which are the fragment ions larger than m/z 43 that are most intense in the CHO1 ion family (Figure 3). On the other hand, Factor 4 is almost

as low as Factor 3 in the more oxidized $f_{44}$ (4 %) and also in O:C ratio (0.34) and $OS_C$ (-1.03) (Table 2).

Generally seen, the factors related to temperature variation (Factors 1 and Factor 4) show a larger difference in oxidation level than the factors related to α-pinene concentration, i.e. particle mass loading (Factors 2 and 3). This suggests that within the investigated conditions, differences in temperature (20 °C to -15 °C) have a larger effect on particle chemical composition than VOC concentration (10 and 50 ppb α-pinene).

decoupled, overall, the four PMF factors represent different main characteristics of the particle chemical composition associated with temperature and VOC precursor concentration, and provide a useful framework for discussing effects of temperature and VOC concentration on SOA formation and properties in chamber experiments. Figure 2 and S7 show that within each experiment, the relative contribution of the factors changes with time. These changes in relative ratios likely reflect

the changes in SOA composition from nucleation (beginning of experiment), condensational growth (increase in mass concentration), and wall loss (decrease in mass concentration towards the end of the experiment). In addition, ongoing gas phase chemistry may also affect observed trends in composition (Kristensen et al., 2020). Furthermore, recent studies (Pospisilova et al. 2020) have shown that particle phase processing continues after condensation, but more work is needed to understand the extent and mechanisms of such processes, and therefore we cannot conclude on such effects.

In each chamber experiment, the correlation between the relative contribution of each factor and the SOA mass concentration can be utilized to infer information about the relative volatilities of the species in each factor. For example, Figures 2 and S7 show that within each of the experiments 1.1 and 1.2, the relative mass concentration ratio of Factor 2 to Factor 1 is largest at lower SOA mass concentrations. This suggests that the volatility of species related to Factor 2 is lower than Factor 1 species

which is interesting since Factor 1 is more oxidized than Factor 2 (Table 2). Figure 2 and S7 show that for each of the experiments 3.2 and 3.3 the trend in the relative mass concentration ratio of Factor 3 to Factor 4 is largest at time periods in the experiment with lower SOA mass concentrations. This indicates that the volatility of Factor 3 species is lower than Factor 4 species. The relative volatilities of Factor 1 and Factor 3 can be assessed by examining the time trends in experiments 3.1 and 3.2 where the fraction of Factor 1 is higher at low mass loading. This together with the fact that the relative ratio of Factor

3 to Factor 1 is higher at lower temperatures suggest that Factor 3 is more volatile than Factor 1. Taken together these results suggest that the volatility (c*) of the four factors increases in the following way: Factor 2 < Factor 1 < Factor 3 < Factor 4.

While we do not see any systematic indications of a specific factor being coupled to relative humidity in our experiments, we cannot rule out that the changes in relative humidity during the experiments might have some impact on SOA composition.

As PMF analysis is traditionally used on ambient data, it is relevant to compare the findings from the AURA chamber experiments to ambient studies. Here the analysis presented by Lee et al. (2016) is relevant, as they used PMF analysis to explore the SOA sources in a coniferous forest mountain region in British Columbia, where SOA concentrations reached up

to 5 µg m$^{-3}$ and the temperature varied from ~5 °C to ~25 °C, corresponding to the temperature in the upper range of the experiments presented in this paper. PMF factors obtained from the ambient AMS data showed a background source and two biogenic SOA sources: BSOA1 from terpene oxidized by ozone and nitrate radical during nighttime, and BSOA2 from terpene oxidized by ozone and OH-radical during daytime. Especially the BSOA1 O:C ratio (0.56), H:C ratio (1.56), and the overall

distribution of peaks (particularly with respect to the relative ratios of m/z 58 to m/z 55 and to the ions above m/z 60) in the mass spectrum, are comparable to Factor 1 (high temperature). Since both ozone and OH-radicals are present in the ACCHA campaign experiments as discussed by Quéléver et al. 2019, is it interesting that the comparability to the Lee et al. BSOA1 factor representing ozone and nitrate radical at nighttime is higher, than the comparability to the BSOA2 factor representing terpene oxidized by ozone and OH-radical at daytime. It may be related to a larger fraction of SOA being formed from

ozonolysis rather than OH-oxidation in the ACCHA campaign experiments (Quéléver et al., 2019). Moreover, it suggests that in Lee et al., during nighttime (BSOA1) ozonolysis might have been the major SOA-forming pathway.

The comparison and similarity between PMF factors from laboratory and ambient observations indicates that the PMF analysis of chamber SOA chemical composition, obtained under different temperature and loading conditions, can be useful for the interpretation and understanding of ambient SOA composition and vice versa.

## 3.2 Trends in chemical composition

### 3.2.1 Elemental analysis

Studying the evolution of the elemental composition of SOA can provide insight into the chemical changes occurring during chemical and physical processes. Figure 4 illustrates the evolution in the O:C ratio during constant temperature experiments

and temperature ramps (standard errors are shown in S25). The time series of the O:C ratio in both 10 ppb α-pinene experiments (1.1-1.5) and 50 ppb α-pinene experiments (2.1-2.3 and 3.1-3.3) are shown. In accordance with expectations based on the absorptive equilibrium framework (equation 1), previous work (Shilling et al. 2009), and observations from the PMF analysis, SOA formed from low α-pinene concentration (10 ppb) and at higher temperature are associated with higher O:C ratios, compared to SOA formed from high α-pinene concentration (50 ppb) at lower temperatures. In all experiments, an initial

increase in the O:C ratios, which subsequently level off, is observed - most significantly in the 50 ppb α-pinene experiments (2.1-2.3) probably due to higher reaction rate in these experiments. Although ageing of oxidized organic particles in ambient measurements is associated with an increase in O:C ratio (Ng et al. 2011) at higher particle mass concentrations, the O:C ratio is usually observed to decrease during particle aging (Shilling et al. 2009; Chhabra et al. 2010; Denjean et al. 2015b), because of increased partitioning of less oxidized semi volatile compounds into the particle phase.

In the 10 ppb α-pinene experiment (1.4), where the temperature is lowered 36 minutes after experimental start, the temperature change from 20 °C to -15 °C is associated with a small decrease in the O:C ratio, which corresponds to condensation of less oxidized, i.e. more volatile, species (Figure 4). Conversely, heating the particles from -15 °C to 20 °C 34 minutes after experimental start (experiment 1.5), results in a slight increase of O:C as more volatile, less oxidized species evaporate and increase the O:C ratio of the remaining particle mass. For comparison, Denjean et al. (2015a) also observed a slight increase

in O:C ratio when increasing the temperature by 6 °C in the room temperature range.

An important outcome of Figure 4 is that the O:C ratios at the end of the temperature ramps are closer to the O:C ratios of the particles in the experiments conducted at the temperature, where the ramps start than where they end. This observation suggests that the composition of α-pinene derived SOA particles is, to a large extent, controlled by the temperature at which they are initially formed, and that subsequent changes in temperature, even as dramatic as 35 °C during 100-130 minutes, only affect

the particle chemical composition to a minor extent. Even though the newly formed particles are exposed to this change in

temperature (~35 minutes after experimental start, and ~1 hour before SOA mass peak (Figure 2)), only slight changes in the chemical composition are observed (Figure 4).

It is also relevant to investigate the corresponding evolution in the O:C ratio and H:C ratio derived from the AMS data based on the so-called Van Krevelen plot (Van Krevelen 1950) (Aiken et al. 2007; Aiken et al. 2008; Canagaratna et al. 2015). Van Krevelen plots of the constant temperature experiments (1.1-1.3, 2.1-2.3, and 3.1-3.3) are shown in Figure 5 (standard errors of O:C ratios and H:C ratios can be found in S25 and S26, respectively). The differences between the 50 ppb α-pinene experiments conducted at similar temperature might be a result of experimental uncertainty. The figures reveal interesting tendencies, both in relation to elemental composition at particle mass peak (Figure 5a) and to evolution during the experiments (Figure 5b). For comparison, the O:C ratio and H:C ratio of the four factors from the PMF analysis, which were obtained from the combined dataset of several experiments, are also shown in Figures 5a and 5b. The O:C ratios and H:C ratios of the factors encompass the data from the individual experiments, illustrating how the factors from the PMF analysis capture and define the extremes in the diversity in chemical composition in particles from the individual experiments, as a result of different experimental conditions. The largest differences in elemental ratios are observed between the SOA particles produced under different temperatures.

Figure 5b shows that during all constant temperature 10 ppb α-pinene experiments, the O:C ratio and the H:C ratio are almost constant. Interestingly, in the 50 ppb α-pinene experiments conducted at lower temperatures (0 °C and -15 °C, experiments 2.2, 2.3, 3.2, and 3.3), the H:C and O:C ratios increase simultaneously during the experiment. As this is not a commonly reported trend, neither in ambient measurements (Ng et al. 2011; Lee et al. 2016) nor in chamber experiments focusing on α-pinene derived SOA (Chhabra et al. 2011) it demonstrates the importance of investigating SOA particles at low, atmospheric relavant, temperatures. Several mechanisms could potentially explain the observed evolution of SOA elemental composition in the Van Krevelen plot, and in fact, it could be due to a combination of different simultaneous mechanisms, e.g. oxidation and oligomerization. Since no OH-scavenger is added in our experiments, one explanation could be related to OH chemistry. Qi et al. (2012) demonstrated that exposure of ozonolysis generated α-pinene SOA, to OH-radicals, increases the O:C ratio, and also leads to higher H:C ratio, because of OH addition to the unsaturated VOC. Modelling suggests, however, that the OH oxidation is not more pronounced at low temperature (0 °C), compared to high temperature (20 °C) (Quéléver et al., 2019), which makes this a less likely explanation for the continuous increase in the O:C ratio and H:C ratio in the cold experiments. More specifically, the ratio of VOC oxidized by ozone relative to that oxidized by OH-radicals was ~2:1, independent of precursor concentration and temperature (Quéléver et al., 2019).

While the simultaneous increase in H:C ratio and O:C ratio could also be associated with hydration reactions (Heald et al. 2010) of carbonyls (Axson et al. 2010), condensation of water does not influence the elemental ratios derived from the AMS spectra, since the calculation does not directly utilize measured $H_2O$-related ion signals, as they typically have large interferences from gas phase $H_2O$ in air (Canagaratna et al. 2015). It should be mentioned, that the observed increase in H:C ratio could potentially be due to impurities condensing to the particle phase in the cold experiments, although this seems highly unlikely, as the chamber was cleaned thoroughly before each experiment (see Kristensen et al. 2020), and the observed changes in the H:C ratios would need an excessive amount of impurities as the particle mass is high (see Table 1, and Figure 2).

The combined effect of the evolution in the O:C ratio and H:C ratio is shown in Figure 5c, depicting the average carbon oxidation state $OS_C$ (Kroll et al. 2011) during the experiments. At all temperatures the 50 ppb α-pinene experiments reach a relatively stable $OS_C$ within ~15 minutes. This suggests that the observed increase in the H:C ratio during the cold 50 ppb α-pinene experiments (Figure 5.b) only has a small effect on the oxidation state of the particles. Throughout the 10 ppb α-pinene experiments conducted at constant temperature (experiments 1.1-1.3) and during heating (1.5), a slight gradual increase in $OS_C$ is observed. For both the 10 and 50 ppb α-pinene experiments, the $OS_C$ is linear correlated with the SOA density (S27).

### 3.2.2 Oxidized organic tracer ions

As described in relation to the mass spectra obtained from the PMF analysis (Figure 3), differences in VOC precursor concentration (i.e., particle mass loading) and temperature primarily result in intensity differences in the dominant oxygen-containing ions, m/z 43 and m/z 44. m/z 43 (dominated by $C_2H_3O^+$ (CHO1 family)) likely derives from organic compounds containing non-acid oxygen (Ng et al. 2010), while the signal at m/z 44 (primary $CO_2^+$ (CHOgt1 family)) arises from carboxylic acids (Alfarra 2004). Both the number of acid groups and the length and functionalization of the carbon chain in the compounds affect the intensity of the signal at m/z 44 (Alfarra et al. 2004; Canagaratna et al. 2015).

Figures 6a and 6b are "triangle plots" (Ng et al. 2010), showing $f_{44}$ (the fraction of m/z 44 relative to the total mass in the spectra) as a function of $f_{43}$ (the fraction of m/z 43 relative to the total mass in the spectra), obtained from unit mass resolution data from the AMS. Figure 6a shows the values at the peak of mass concentration (five data point average) of constant temperature experiments (1.1-1.3, 2.1-2.3, and 3.1-3.3) and temperature ramp experiments (1.4 and 1.5), while Figure 6b shows the evolution through the constant temperature experiments. As observed in the Van Krevelen plots (Figures 5a and 5b), data from the repeated 50 ppb α-pinene experiments (2.1-2.3 and 3.1-3.3) conducted at similar temperatures show overall reproducibility, though not being identical.

The triangle plots show that particles formed at higher temperature have a higher $f_{44}$ (i.e. $CO_2^+$, acid-derived functionalities) than particles formed at lower temperature. No clear trends with temperature are observed for $f_{43}$ (i.e. $C_2H_3O^+$, non-acid-derived functionalities). Particles formed at lower α-pinene concentration (10 ppb) have higher $f_{44}$ and a lower $f_{43}$ than particles formed at higher α-pinene concentration (50 ppb). This suggests that acid-derived functionalities are more prevalent in α-pinene SOA formed at low precursor concentration (and thus low particle mass loading), which is consistent with less partitioning of the more volatile, less oxidized material to the particle phase. In all experiments, $f_{44}$ values are between 0.04 and 0.1, and $f_{43}$ values are between 0.08 and 0.15. These levels are comparable to values reported in the literature from chamber experiments conducted at comparable α-pinene concentrations, both at room temperature (Chhabra et al. 2011; Kristensen et al. 2017) and at -15 °C (Kristensen et al. 2017). The slight continuous increase in $f_{43}$ (Figure 6b) in the experiments conducted at -15 °C (experiments 1.3, 2.3, and 3.3) is in agreement with observations by Kristensen et al. (2017) from experiments performed at -15 °C at identical α-pinene concentrations. Moreover, the non-evolving $f_{43}$ at 20 °C is also in agreement with literature exploring α-pinene SOA at comparable concentrations and room temperature (Chhabra et al. 2011; Kristensen et al. 2017). As the increase in $f_{43}$ is only observed in the cold experiments, especially -15 °C, this suggests that formation of species that give rise to high $f_{43}$ values are highly temperature dependent.

### 3.2.3 Estimated particle content of organic acids

In AMS mass spectra, m/z 44 has been shown to be a good tracer for the content of organic acids in SOA (Canagaratna et al. 2015; Yatavelli et al. 2015). Yatavelli et al. (2015) investigated how the mass concentration of molecules (R-COOH) containing one or more acid functionalities, can be related to the AMS derived mass concentration of m/z 44 multiplied by scaling factors. Yatavelli et al. (2015) estimated that 10 to 50 % of the organic particle mass in the northern hemisphere can be attributed to molecules containing the carboxylic acid functionality. Inspired by Yatavelli et al. (2015), we here explore how the intensity of m/z 44 in the AMS mass spectra compares to the mass concentration of organic acids (R-COOH) and organic acid functionalities (-COOH) based on results from the off-line LC-MS analysis of filter samples obtained by the end of the AURA chamber experiments. As described in detail in Kristensen et al. (2020), LC-MS analysis was performed to identify and quantify 10 carboxylic acids formed in the dark ozonolysis of α-pinene (constituting 18-38 % of the SOA mass concentration in current experiments), as well as 30 dimer esters (constituting in total 4-11 % of the total SOA mass concentration in current experiments).

For the 50 ppb α-pinene experiments 3.1, 3.2, and 3.3, conducted at 20 °C, 0 °C, and -15 °C, respectively, Figure 7a shows the mass concentration of organic acids (R-COOH) identified from LC-MS analysis (Kristensen et al., 2020), and the mass concentration of the m/z 44 signal in the AMS mass spectra, scaled to the SMPS mass concentration and corrected for density, as previously described. For both techniques (AMS and LC-MS), the mass concentration of organic acids is lower at higher

temperatures. The AMS m/z 44 mass concentrations are lower than the organic acid concentrations obtained from the LC-MS by factors of 2.55, 4.11, and 4.65 at 20 °C, 0 °C, and -15 °C, respectively. In the following, these numbers will be referred to as scaling factors. For comparison, Yatavelli et al. (2015) reported the m/z 44 AMS signal being a factor of ~2.32 lower than the mass concentration of organic acids in SOA during summertime in a forest area dominated by pine trees near Colorado Springs, USA. Their result is in very good agreement with the scaling factor obtained in the experiment conducted at 20 °C,

which supports the hypothesis that the most important organic acids in α-pinene SOA are determined by the LC-MS method. The variation in scaling factors at the different temperatures likely reflects that organic acids with different numbers of acid functionality (-COOH) and/or different multifunctional moieties exhibit different degrees of thermal decomposition to the m/z 44 signal in the AMS (Canagaratna et al. 2015; Yatavelli et al. 2015). The similarity of the scaling factors obtained in the 0 °C (4.11) and -15 °C (4.65) experiments is consistent with the fact that the SOA chemical composition at those temperatures have

a higher degree of comparability relative to the 20 °C experiment, where a lower scaling factor (2.55) is obtained (recall Figures 4 and 6).

Since lower SOA mass is produced at the higher temperatures, it is also relevant to investigate how the mass fractions of organic acids vary with temperature (Figure 7b). The mass fractions are obtained by dividing the LC-MS and AMS results presented in Figure 7a by the total SOA mass concentration measured in the chamber, prior to the filter sampling, and corrected

as described previously. By application of the scaling factors found above, the two techniques are in good agreement, though slight differences appears at 20 °C and -15 °C. While the mass concentration of organic acids (R-COOH) obtained from the LC-MS decreased significantly with higher temperature (Figure 7a), no trend is observed in organic acid mass fractions (Figure 7b). Interestingly, for m/z 44 from the AMS mass spectra, the temperature dependent trend changes from decreasing with higher temperature (Figure 7a) to increasing when focusing at the mass fraction of m/z 44 to total SOA mass concentration

(Figure 7b).

Some of the organic acids as well as the dimers observed from the LC-MS data (Figures 7a and 7b) contain multiple acid functionalities (-COOH) (Kristensen et al., 2020). Therefore, it is also relevant to investigate how the mass concentration and mass fraction of acid functionalities (from the suggested molecular structures (Kristensen et al., 2020)) relate to the m/z 44 signal obtained from the AMS. Lower masses of organic acid functionalities (-COOH) are obtained at higher temperatures,

and the scaling factors of 1.15, 1.46, and 1.70 applied to the m/z 44 AMS signal at 20 °C, 0 °C, and -15 °C, respectively (Figure 7c), are lower and less variable with temperature than those for the mass concentrations of organic acids (R-COOH, Figure 7a). The difference between the scaling factors, related to the mass concentration of organic acids (R-COOH, Figure 7a) and organic acid functionalities (-COOH, Figure 7c) at the same temperatures, reflects the mass of the organic acid backbone (i.e. R in R-COOH). The observed trends in Figures 7a and 7c suggest that organic acids with heavier backbones are formed at

temperatures below 20 °C.

Figure 7d shows that carboxylic acid functionalities (-COOH) account for a greater fraction of the observed SOA mass concentration at higher temperatures, consistent with the observation of higher $f_{44}$ values at higher temperatures in Figure 6. This trend is opposite to the temperature dependent trends of the absolute mass concentration of organic acid functionalities (-COOH, Figure 7c) and organic acids (R-COOH), both in this study (Figure 7b) and in previous studies (Zhang et al. 2015;

Kristensen et al. 2015).

Overall, the comparison of the m/z 44 signal from the AMS mass spectra and SOA acid content obtained from LC-MS data shows that organic acids and organic acid functionalities are important constituents of α-pinene derived SOA, and that it is relevant to investigate and compare different techniques for their quantification.

**4 Conclusion**

The chemical composition of α-pinene derived SOA was investigated using HR-ToF-AMS in a series of experiments performed at different α-pinene concentrations (10 ppb and 50 ppb) and temperatures (20, 0, and -15 °C, and ramps in the range 20 to -15 °C and -15 to 20 °C). PMF analysis was applied to a combined AMS dataset representing eight different experimental conditions. The PMF analysis revealed that the chemical composition of the SOA particles could be described by four factors, which differ in their dependence on VOC concentration and experiment temperature. To our knowledge, this is the first study using PMF analysis on AMS chamber data to reveal distinct factors sensitive to temperature.

This analysis demonstrates that α-pinene SOA oxidation level is dependent on both temperature and α-pinene concentration: SOA oxidation level increases with higher temperature and with lower SOA mass loading. The dataset suggests that particles formed at 0 °C are more chemically similar to particles formed at -15 °C than to particles formed at 20 °C. Temperature ramps over a range of 35 °C were only accompanied by slight changes in chemical composition, with increasing oxidation levels during heating ramps and decreasing oxidation levels during cooling-ramps. The investigation demonstrates that the temperature at which particles are formed is decisive for their properties during α-pinene SOA lifetime. This is interesting from an atmospheric perspective as secondary organic aerosol particles are formed and age over a wide range of temperatures.

**Acknowledgements**

This work was supported by Aarhus University, The Aarhus University Research Foundation (AUFF), The European Research Council (ERC-Grant nr: 638703-COALA), and the Academy of Finland, Centre of Excellence program (project nr:307331).

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

**Table 1**. Overview of experimental details for the ACCHA experiments included in this work. The experiments are constant temperature (const. temp.) experiments and/or temperature ramp (temp. ramp) experiments.

| Exp. # | Exp. ID | Exp. type | O₃ | α-pinene | Temp. avg. at const. temp. (±std. dev.)* | RH avg. at const. temp. (±std. dev.) [start to end]* | Ramp start-to-end temp | Ramp start-to-end RH | Density |
|---|---|---|---|---|---|---|---|---|---|
| | | | ppb | ppb | °C | % | °C | % | g cm⁻³ |
| 1.1 | 20161202 | Const. temp (20 °C) | 104 | 10 | 20.2 (±0.0) | 0.5 (±0.7) [0.0-2.1] | | | 1.25 |
| 1.2 | 20161208 | Const. temp (0 °C) | 105 | 10 | 0.0 (±0.2) | 6.0 (±2.9) [2.8-12.7] | | | 1.20 |
| 1.3 | 20161207 | Const. temp (-15 °C) | 106 | 10 | -15.1 (±0.3) | 10.3 (±2.1) [8.0-15.0] | | | 1.16 |
| 1.4 | 20161209 | Temp. ramp (20 to -15 °C) | 103 | 10 | | | 19.8 to -12.0 | Na | 1.23 |
| 1.5 | 20161220 | Temp. ramp (-15 to 20 °C) | 113 | 10 | | | -14.0 to 19.6 | 11.7 to 1.7 | 1.17 |
| 2.1 | 20161212 | Const. temp (20 °C) | 105 | 50 | 20.0 (±0.2) | 0.8 (±0.9) [0.0-3.0] | | | 1.23 |
| 2.2 | 20161219 | Const. temp (0 °C) | 107 | 50 | -0.3[A] (±0.1) | 7.0[A] (±0.2) [6.9-7.5] | | | 1.15 |
| 2.3 | 20161221 | Const. temp (-15 °C) | 113 | 50 | -15.0 (±0.2) | 24.7 (±3.6) [19.3-31.7] | | | 1.12 |
| 3.1 | 20170112 | Const. temp (20 °C) | 100 | 50 | 20.0 (±0.2) | 1.7 (±1.6) [0.0-4.6] | | | 1.21 |
| 3.2 | 20170116 | Const. temp (0 °C) | 105 | 50 | 0.0[B] (±0.1) | 8.9[B] (±0.1) [8.6-9.1] | | | 1.14 |
| 3.3 | 20170113 | Const. temp (-15 °C) | 105 | 50 | -15.4 (±0.6) | 14.5 (±3.8) [11.1-23.0] | | | 1.13 |

[A]Temperature and relative humidity probe failure after 59 minutes and the rest of the constant temperature experiment; [B]temperature and relative humidity probe failure after 42 minutes; *averages of temperature and relative humidity are based on data from the time of α-pinene injection to filter sampling and might differ slightly from the values reported in Kristensen et al. 2020. Densities are 30 minutes averages obtained by the end of the experiments (1.1-1.5 and 3.1-3.3).

**Table 2.** Characteristics of the four factors obtained from PMF analysis of experiments (1.1, 1.2, 1.3, 1.4, 1.5, 3.1, 3.2, and 3.3) according to Figure 2. For each factor, the fraction of m/z 43 ($f_{43}$) and m/z 44 ($f_{44}$) relative to the total mass spectra are stated in percent.

| Factor | Dominating appearance | Characteristics | $f_{43}$ | $f_{44}$ | O:C ratio | H:C ratio | $OS_C$ |
|--------|----------------------|-----------------|----------|----------|-----------|-----------|--------|
| Factor 1 | High temperature | Fraction decrease with decreasing temperature at both α-pinene concentrations. | 14 | 9 | 0.56 | 1.65 | -0.53 |
| Factor 2 | 10 ppb (low) α-pinene concentration, low particle mass loading | High fraction in all 10 ppb experiments. Almost nonexistent in 50 ppb experiments. | 7 | 8 | 0.39 | 1.59 | -0.81 |
| Factor 3 | 50 ppb (high) α-pinene concentration, high particle mass loading | High fraction in all 50 ppb experiments. Appears only slightly in 10 ppb experiments. | 10 | 3 | 0.26 | 1.60 | -1.08 |
| Factor 4 | Low temperature | Fraction decrease with increasing temperature at both α-pinene concentrations. | 13 | 4 | 0.34 | 1.71 | -1.03 |

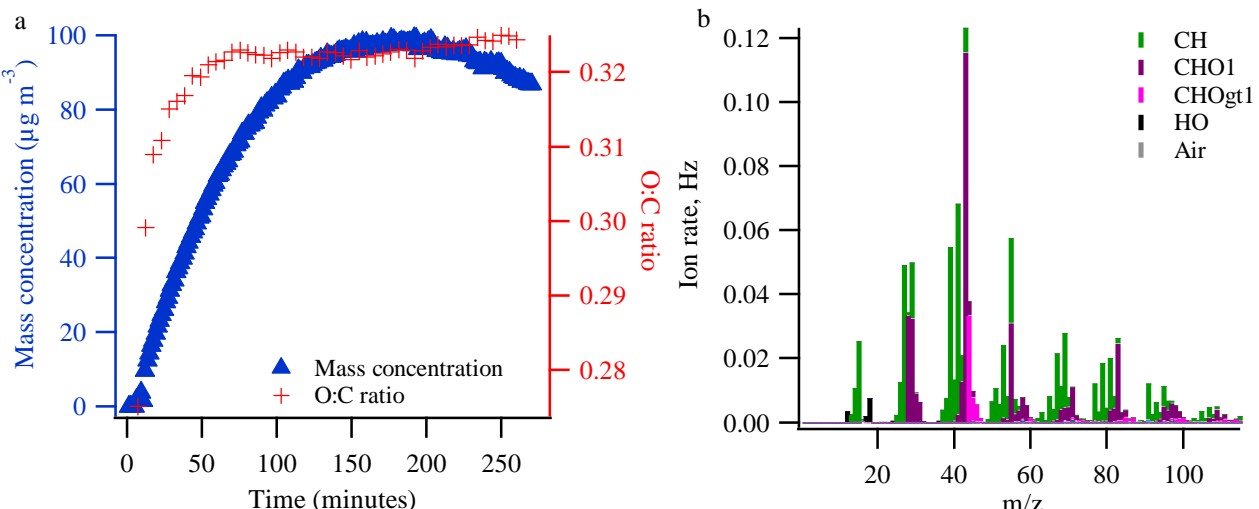

10   **Figure 1**. (a) Evolution of SOA mass (density corrected) and O:C ratio during a typical experiment (50 ppb α-pinene, -15 °C, experiment 2.3) and (b) mass spectra of the experiment obtained at mass peak (5 minutes average). gt means greater than.

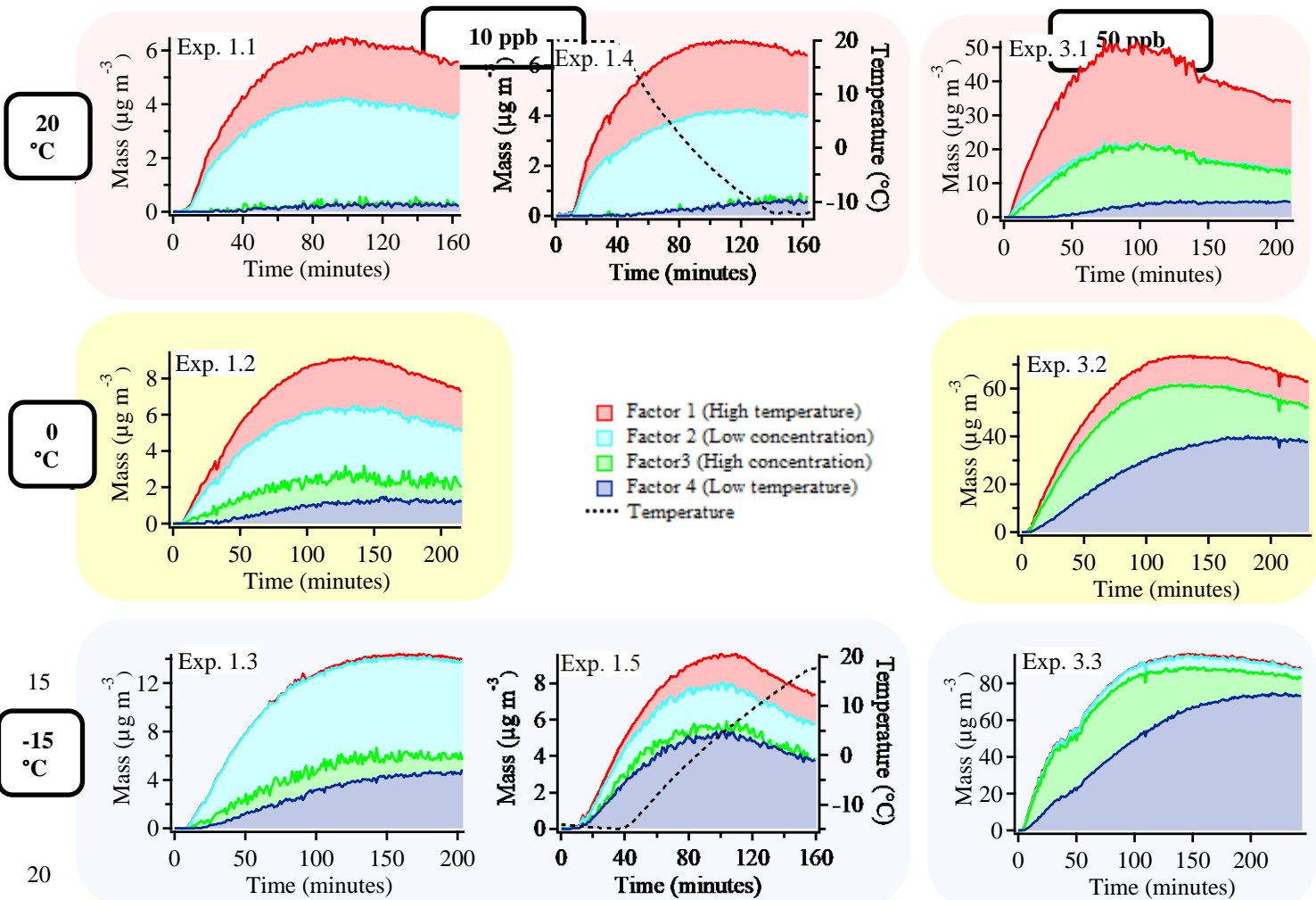

**Figure 2**. Mass evolution (µg m$^{-3}$) of the four factors from PMF analysis of the combined AMS data set including six constant temperature experiments with an initial α-pinene concentration of 10 ppb or 50 ppb, conducted at 20 °C (experiments 1.1 and 3.1), 0 °C (experiments 1.2 and 3.2), and -15 °C (experiments 1.3 and 3.3) and two temperature ramp experiments with an initial α-pinene concentration of 10 ppb where the temperature was changed from 20 °C to -15 °C (experiment 1.4) and from -15 °C to 20 °C (experiment 1.5). Graphs in the left and middle column depict 10 ppb α-pinene experiments and 50 ppb α-pinene experiments are shown in the right column. Each row of graphs represent different initial temperatures. The reason for the bumb/shoulder around 50 minutes in experiment 3.3. is unclear.

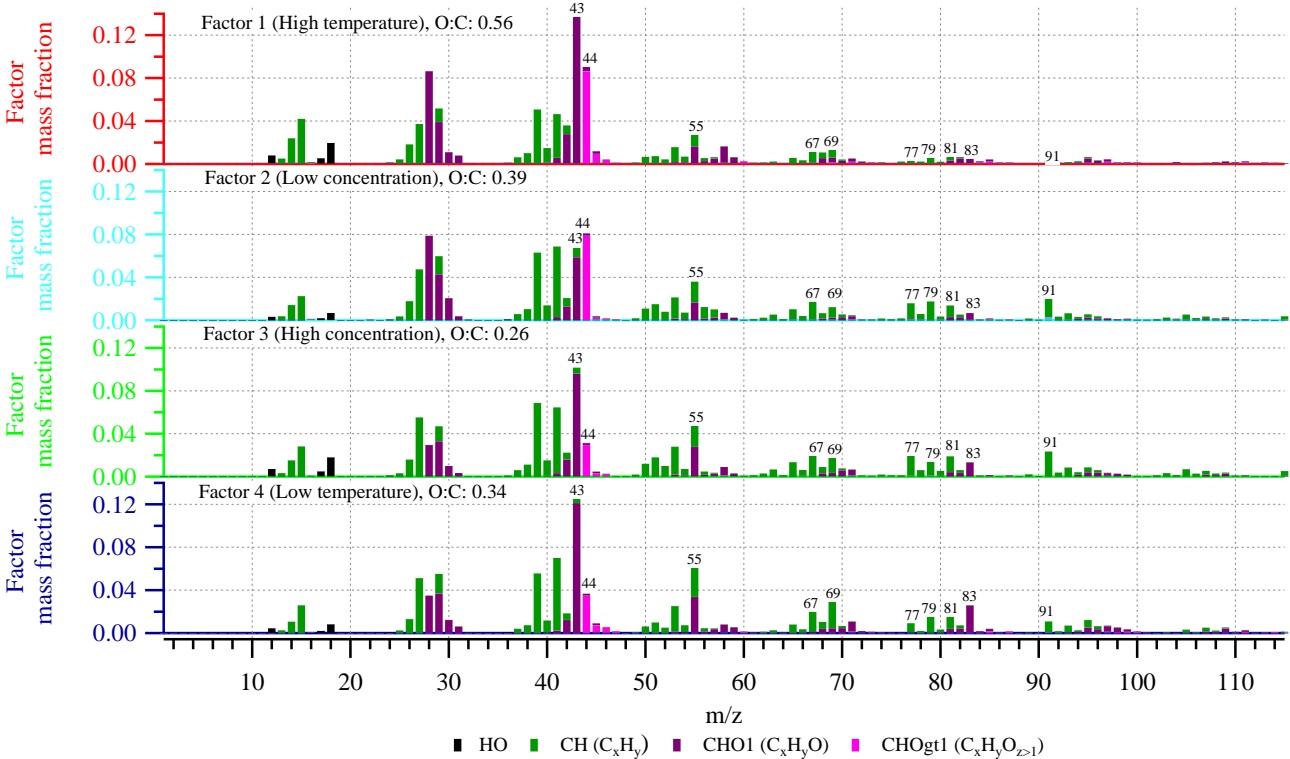

**Figure 3.** Mass spectra of the four factors from the PMF analysis (see also Figure 2 and Table 2) of the combined data set of experiments 1.1-1.5 and 3.1-3.3. gt means greater than.

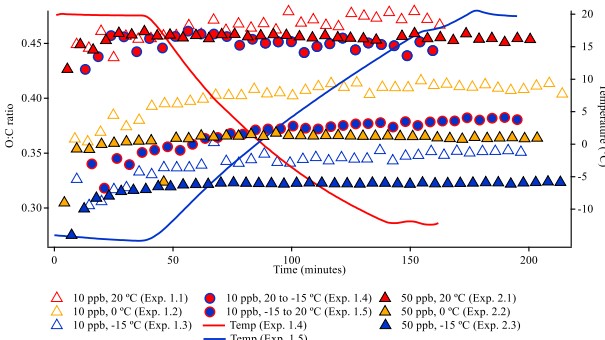

**Figure 4.** Time series of O:C ratio in 10 ppb α-pinene experiments (1.1-1.5) and 50 ppb α-pinene experiments (2.1-2.3). Every fifth data point is shown. Standard errors are shown in S25.

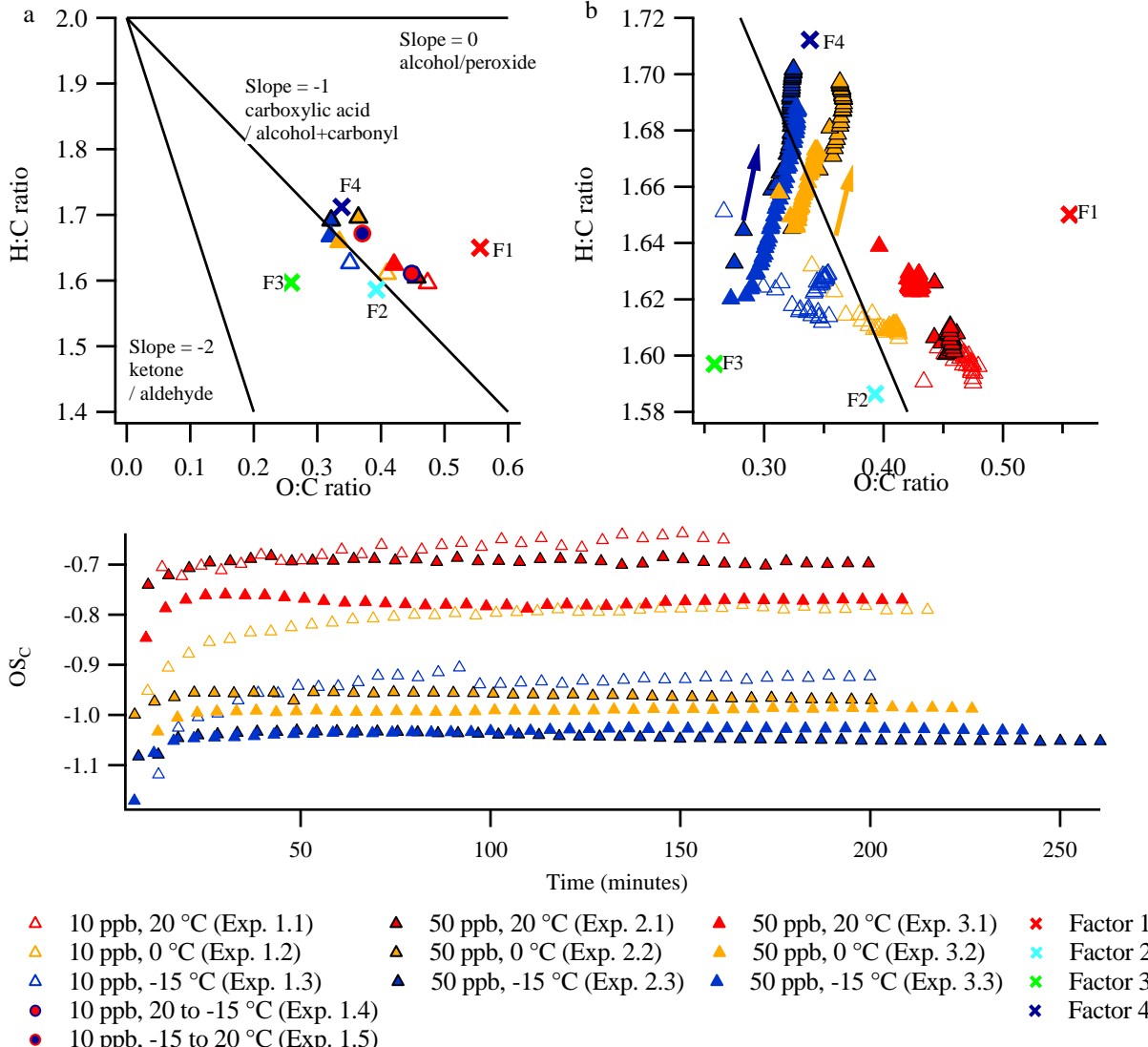

**Figure 5.** Van Krevelen plots (H:C ratio vs. O:C ratio) at (a) SOA mass peak (five data point average) and in the expansion (b) during the constant temperature experiments in this work (arrows indicate direction of time), as well as (c) the $OS_C$ during the experiments. The lines in the Van Krevelen plot are based on Heald et al. (2010) and Ng et al. (2011). Notice the different scales of the axes in panels a and b. The positions of the four factors obtained from the PMF analysis are indicated by crosses: F1: Factor 1 (high temperature), F2: factor 2 (low concentration), F3: Factor 3 (high concentration), and F4: Factor 4 (low temperature). Panels b and c show every fifth data point. Standard errors are shown in S25 and S26.

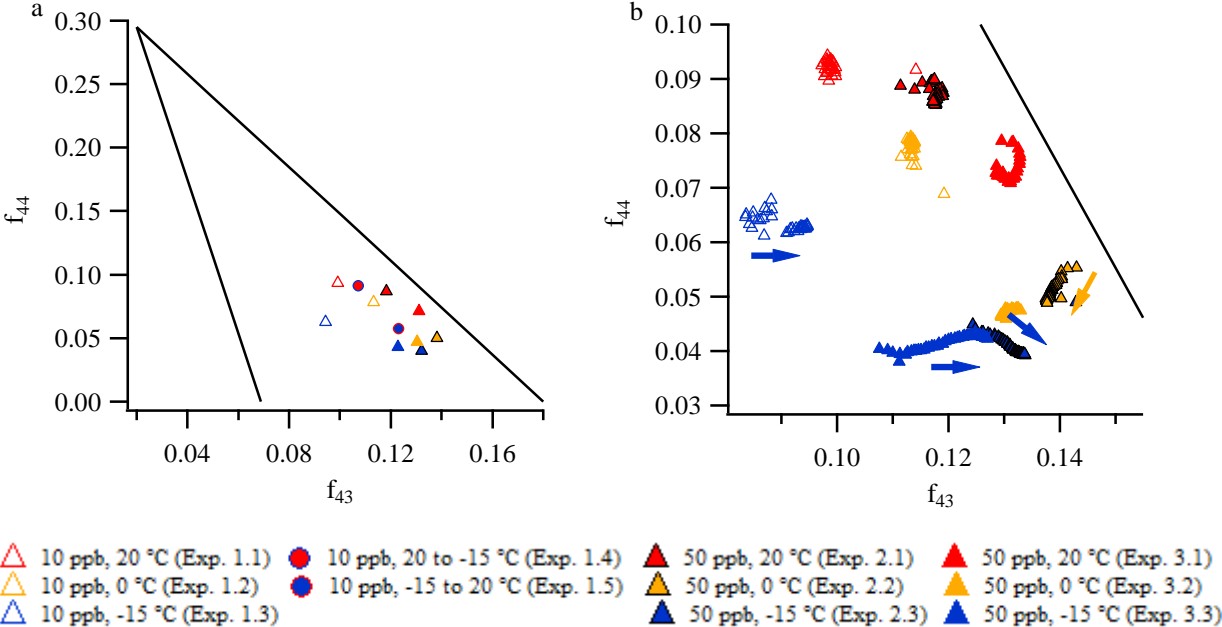

**Figure 6.** The elemental composition of particles formed in experiments 1.1-1.3, 2.1-2.3, and 3.1-3.3 depicted in triangle plots ($f_{44}$ vs. $f_{43}$) (a) at mass peak (five data point average) and in the expansion (b) during the experiments (~5 minutes time resolution, arrows indicates direction of time). Notice the different scales of the axes in panels a and b. In the triangle plots the lines ($y = -6.0204x + 0.4154$ and $y = -1.8438x+0.3319$, where $0.069 \leq x \leq 0.18$ and $y \leq 0.295$) define the common composition of oxygenated organic aerosol (Ng et al., 2010).

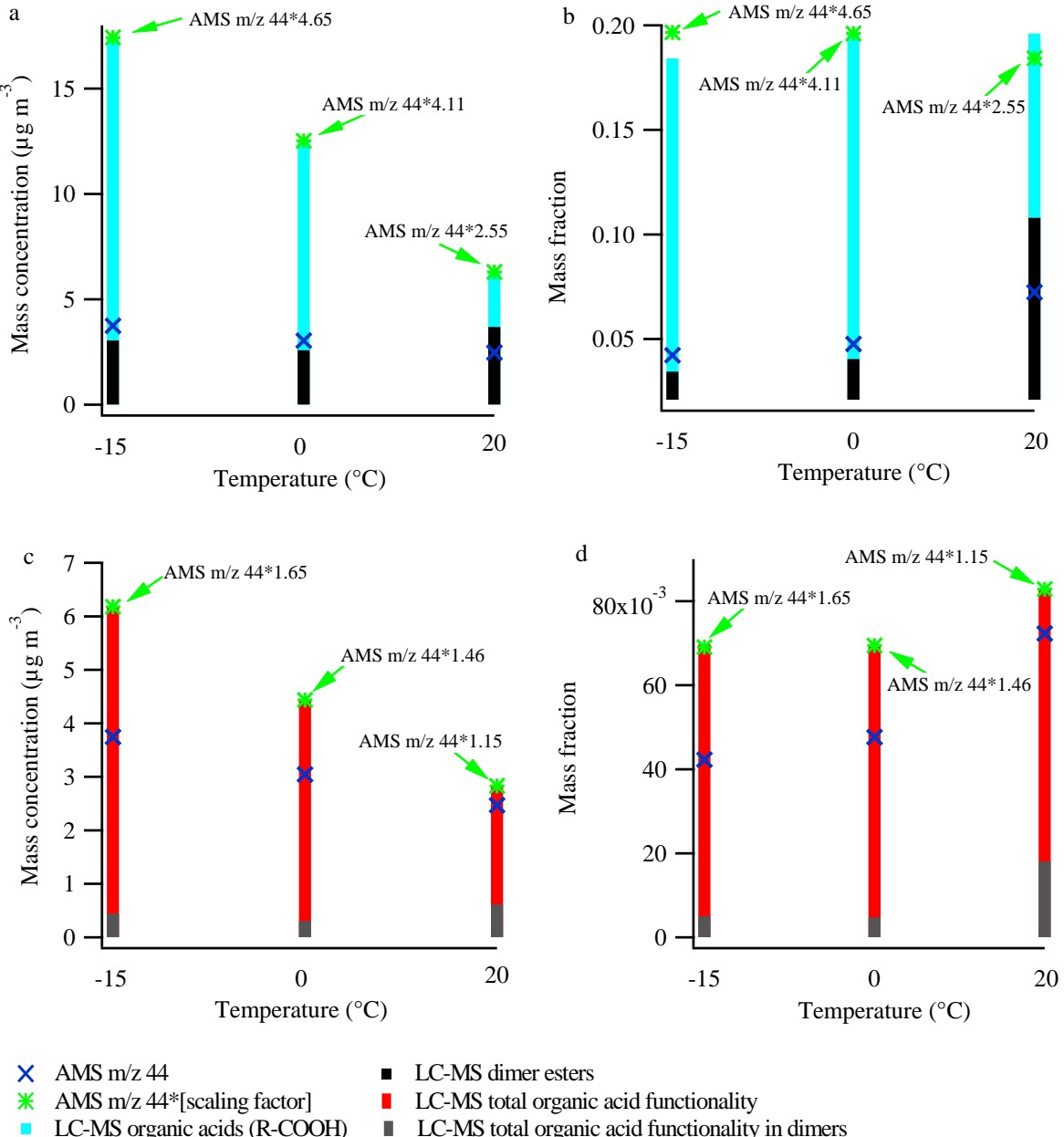

✕  AMS m/z 44                          ■  LC-MS dimer esters
✳  AMS m/z 44*[scaling factor]         ■  LC-MS total organic acid functionality
■  LC-MS organic acids (R-COOH)        ■  LC-MS total organic acid functionality in dimers

**Figure 7.** LC-MS derived organic acid (R-COOH) mass concentration (panel a) and organic acid mass fraction (panel b) at -15, 0, and 20 °C. The organic acid functionality (-COOH) mass concentration and mass concentration at the three temperatures are shown in panels c and d respectively. Additionally the corresponding results related to dimers are shown. The panels also show m/z 44 - a tracer of organic acids - from the AMS mass spectra and the scaling factors to be applied to reach the level of organic acid (functionality) measured by the LC-MS. The particle filter samples analyzed by the LC-MS are obtained by the end of the experiments (see Table 1) while the AMS results are obtained right before the filter sampling, 10 minutes average). The figure is based on data from experiments 3.1-3.3.