# Peer review of "Temperature and VOC concentration as controlling factors for chemical composition of alpha-pinene derived secondary organic aerosol"

_Atmospheric Chemistry and Physics, 2020_

## Referee Comment (RC1) · Anonymous Referee #1 · 6 Apr 2020

Review of Jensen et al.: Temperature and VOC concentrations as controlling factors for chemical composition of alpha-pinene derived secondary organic aerosol

Summary This paper investigates the effects of temperature and alpha-pinene concentration on the chemical composition of SOA particles formed via dark ozonolysis in a series of chamber studies. Experiments were performed in the dark at two concentrations, 10 ppb and 50 ppb, in the presence of ozone. Three constant temperatures were analyzed (20, 0, and -15 C), and several temperature ramps were induced at various points during the ozonolysis. Chemical composition was investigated in real-time using an HR-ToF-AMS, and offline chemical composition of filter samples was performed via

LC-MS. The authors apply PMF to the AMS data in order to determine controlling factors behind the chemical composition. They discuss a four-factor solution, where two of the factors correspond to temperature and two factors correspond to alpha-pinene concentration. The authors argue that the temperature at which particles initially form is more important for determining particle composition than changes in temperature after the initial formation. However, they do find that temperature ramps lead to slight changes in oxidation levels. The authors also compare AMS derived estimates of organic acid content to off-line LC-MS analysis of filters and find good agreement. They conclude that high temperatures and low initial alpha-pinene concentrations are associated with more oxidized SOA.

General Comments This paper is not suitable for publication in its present form. In general, a more rigorous and thorough discussion of results with consideration of context would be helpful. There is a tendency in this manuscript to state results and not interpret them. I also take issue with some of the findings.

The second to last sentence of this manuscript is "This work confirms that the particle chemical composition is dependent on precursor concentration and particle mass loading." This is not explored in the manuscript.

The authors state that "Factor 1" is a temperature factor, however this does not seem to be the case. Comparing experiment 1.4 to 3.1 (same constant temperature) Factor 1 appears to change more than a factor of 10 for a factor of 5 change in precursor VOC concentration. Further, Factor 1 does not change during the lower VOC (10ppb) temperature ramp experiments. Factor 1 is essentially the same for experiments 1.4 vs. 1.5, though this is a little hard to discern because the y-axes are different. The authors admit that a five- or six-factor solution provides a better explanation for the data than the four-factor solution presented in the manuscript, but do not adequately explore why the higher-order factor solutions are discarded. Ramp start-to-end RH is presented in Table 1, but it is not explored in the manuscript. RH values change drastically during the temperature ramp, but this information is not used in the analysis. Is it possible

that the ignored factors from the PMF solution are instead factors relating to RH? If so, they should be included in the analysis and their importance should be discussed, or a more adequate reason for discarding them should be addressed.

Atmospheric Chemistry and Physics is an applied journal and it surprises me the authors provide no atmospherically relevant context for the selected temperatures, initial hydrocarbon concentrations, etc. The authors state early in their manuscript that oxidation levels of SOA from a-pinene ozonlysis decrease with increased particle mass loadings. The authors' findings do not seem to be consistent with this. Why?

Specific comments and line numbers follow: Page 2, Line 6: SOA is also formed through the uptake of water-soluble organic gases into condensed-phase water.

Page 2, Line 7: The 1995 Guenther manuscript describing a biogenic emissions model is not the right reference for "SOA forms from biogenic and anthropogenic precursors".

Page 2, Line 10: Hydroxyl radical (OH) should have a dot, not a minus sign.

Page 3, Line 28-32: Why were different precursor concentrations chosen for the temperature-ramp experiments? If precursor concentration is playing an important role in chemical composition, then why aren't all the temperature-ramp experiments performed using the same initial alpha-pinene concentration? Here, some are done with 10 ppb and some with 50 ppb. Why are these conditions relevant?

Page 4, Line 36 – Page 5, Line 1: According to Table 1, these experiments also saw a wide range of RH, which could lead to the condensation of water onto the particles and subsequent aqueous-phase reactions. Is it possible that Factors 5 and 6 are related to RH? If so, please address this in the manuscript, or provide a justification for why these factors and the large range of RH values can be ignored.

Page 5, starting at Line 14: The authors state, "Under all conditions (i.e. in all experiments), the AMS derived SOA densities are in the range of 1.1 to 1.3 g cm-3. Figure 1 indicates that SOA density is < 1.1 for the first 0.5 hour.

Page 5, Line 32-35: Please provide a reasoning for using only some of the temperature ramp experiments in the PMF analysis.

Page 6, Line 29-30: What contributions from f44 and m/z ratios do Factors 5 and 6 have?

Page 7, Line 12-14: Please provide a discussion regarding why it makes sense to have less-oxidized SOA at low temperatures and high alpha-pinene concentrations. It is interesting that this is the case, but a scientific justification is missing and should be discussed. Is this a matter of kinetics and competition for oxidants?

Page 7, Line 31: Why is no OH scavenger used, and how does potentially changing [OH] confound interpretation of O:C ratios and factors? The authors provide some speculation on Page 7, but the context is insufficient.

Page 8, Line 18-24: A lot of time and energy is spent on discussing the reasons behind changes in the O:C and H:C ratios during temperature ramps in previous paragraphs. However, here it is stated that the changes are small and not very important. If that is the case, then why discuss the changes to the O:C and H:C ratios so extensively? It seems this section could be more focused.

Page 8, Line 39 – Page 9, Line 9: Please provide a reasoning for why a lower precursor concentration leads to more highly oxidized SOA. Is this because there is less competition for oxidants?

Page 10, Line 9-12: The authors state that organic acids with heavier backbones are formed at lower temperatures. Why does this make sense? Is this due to volatility reasons?

Figures and Tables: Table 1: Do I understand correctly that no experiment was repeated and there is no demonstrated repeatability?

Figure 1: Particle density is presented with four significant digits and no estimate of uncertainty. There should be uncertainty bars or some other representation.

Figure 4: In panel b, what is the black line? It is not labeled.

Figure 6: In panel b, the black line is not defined. Why is the black line in panel b different from panel a, if they are both based on Heald and Ng? In panel a, it is unclear how this figure supports the author's conclusions regarding the oxidation level of SOA in each experiment.

Supplemental Information: Figures S5 and S7. Could these extra factors be attributable to RH? They seem to play an consequential role in mass loading.

---

## Referee Comment (RC2) · Anonymous Referee #2 · 24 Apr 2020

Summary:

The authors present a very interesting dataset of secondary organic aerosol (SOA) composition measurements with a High-resolution Time of flight AMS (short: AMS) and offline filters. They investigated the evolution of a-pinene SOA generated via dark ozonolysis in a temperature-controlled chamber either at constant temperatures (20 °C, 0 °C, and -15 °C) or with controlled temperature ramps. They compared the end point composition with offline filter measurements and apply Positive Matrix Factorisation (PMF) analysis to the AMS data. They find PMF factors describing the difference observed in composition due to the changing temperature and factors that capture

the impact of different precursor concentrations. They conclude from the temperature ramp experiments that the temperature at which the particle formation is more important for the overall particle composition than the final temperature. I think this is a great data set with the potential to give important insights into how temperature and precursor concentration control the chemical composition of a-pinene SOA particles which is important to understand when comparing chamber experiments to ambient measurements. Unfortunately, the presented analysis and interpretation of this data set are somewhat superficial. The authors mostly just describe the data but leave the reader wondering about the implications. I therefore request major revisions of the data interpretation and discussion part in addition to addressing my comments below before this paper can be published in ACP.

Major Comments:

1) One reason the discussion of the results is so unsatisfactory is that the authors do not use the additional information available for these experiments. As one example, take the only small changes in O:C observed in the temperature ramp experiments 2.1 - 2.3. How much did the SOA mass concentration / particle size change in that time? I.e., how much was the existing SOA composition perturbed by the T ramp? With that in mind, how much change can be expected in a value like O:C? I.e., if cooling from 20 °C to -15 °C leads to the condensation of x% of compounds with lower O:C how much does that change the O:C of the original particles? Is O:C sensitive enough to monitor such changes? This is just one example of how the authors can improve the interpretation and explanation of their results. I request that the authors utilise the other available information (part of it is already published in Kristensen et al. (2020) and Quelever et al. (2019)) to support their findings and strengthen the discussion.

2) A second reason is that the authors do not use the results of their presented data set to the full extent possible. They conducted PMF analysis and concluded that the grouping of the factors is connected to the SOA formation temperature. Why do they not use this to investigate composition changes in the ramp experiments (Exp 1.5&1.6,

2.1-2.3)? The PMF chapter (3.1) is mostly describing the marker ions in the different factors and comparing the factors with each other (and one ambient example). But there is no analysis of the temporal evolution of the contribution of the factors during the experiments. Chapter 3.2 is instead focusing on the changes in O:C and the two tracer ions m/z 43 and m/z 44. Kristensen et al. (2017) already covers the temperature dependence of the acid and ester dimer formation in detail. The fact that the AMS f44 values agrees with the LC-MS results is nice. I request a more thorough interpretation of the changes in PMF factor contributions during the experiments which will hopefully reveal the "bigger" picture of the composition changes with temperature and precursor concentration as the PMF analysis utilises all ions detected in the AMS and not just m/z 43 and m/z 44.

3) The paper presents many (factor) mass spectra and compares them stating that they are similar (or not). But the authors do not explain what metric they use to determine similarity (statistical method, ratios of marker ions, "by eye"). If they have not used a objective (mathematical) metric, I recommend using the spectral contrast angle, theta, which is derived from the dot product of two mass spectra, A and B (Wan et al., 2002). A and B are considered similar if theta is between 0° and 15°, somewhat similar but with important differences if theta is between 15° and 30°, and different with theta values >30° (Bougiatioti et al., 2014). This parameter (sometimes as cos theta with values between 0 and 1) has been used for AMS data before (e.g. Bruns et al., 2015; Kostenidou et al., 2009). Such a pairwise comparisons will also help with claims like that the composition of the 0 °C and -15 °C experiments are closer to each other than to the 20 °C case. I request that the authors specify how they determined similarity of mass spectra and provide values/graphs that show the degrees of similarity (can be in SI material).

4) The explanation and labelling of the PMF factors as high/low concentration or temperature is a bit misleading and easily misunderstood. Their labelling/description makes it sound as if the effect of higher precursor concentration is covered by switching

from Factor 3 to Factor 2. But the higher cOA (organic aerosol particle mass) also impacts the contribution of the other factors. As an example, Factor 4 (low T) increases its contribution at all temperatures when the VOC concentration is increased. In a similar way, the contribution of Factor 2 (high concentration) strongly depends on temperature (e.g. increasing for 10ppb cases with decreasing temperature). They authors do explain the connection of partitioning with C* and cOA in the introduction in general, but they do not clearly state how the factors fit into this framework. The authors need to clarify how and why the factors are related to cOA and temperature, i.e., interpret the factors with regard of volatility.

5) Continuing the point from the previous comment, from the limited explanations provided in the text, I derive that the authors made the assumption that the factorisation in PMF depends on the condensation behaviour of the compounds which is governed by the temperature and the increasing cOA. I.e., the increase in contribution of factor 4 with time in Exp 3.2 and 3.3 would thus be explained by higher volatility material condensing as cOA increases. But why would then Factor 2 decrease already before the peak in mass concentration is reached? Could that not be connected to ongoing chemical processes? In this paper, there is no information about the consumption of the precursor and thus on how long reactions may be ongoing. How do the different reaction rates at low/high temperature and/or different precursor concentrations affect the oxidation products? This is especially interesting for the T ramp experiments (1.5 & 1.6). The authors have to at least mention if or how ongoing chemistry would impact the PMF factors and volatility of reaction products. If the authors merely want to compare the final composition of the formed particles, they need to clarify that and provide information how they determine the "end point".

6) As in many papers with AMS data, the authors use the O:C values as a parameter for the oxidation level of the investigated SOA particles. However, if the H:C values also change (as they point out when discussing the van Krevelen diagram in Fig 4b), the case is not that simple and the average oxidation state of carbon (OSC, Kroll et

al. (2011)) may be the better parameter representing the degree of oxidation in the particles. As an example, for Exp 3.3 (50 ppb, -15 C) in Fig 4b, I read a change for O:C & H:C values from 0.28 & 1.62 to 0.33 & 1.705, respectively. Calculating OSC as OSC = 2*O:C – H:C gives 1.05 and -1.06 for the start and end of the experiment, i.e., the average oxidation state of carbon did not change. As another example, the OSC values of factors 2 and 4 are almost identical although the O:C values differ. Do the authors have a reason why O:C would be a better measure of the level of oxidation in their experiments than OSC? If not, they should change to using OSC to account for changes both in O:C and H:C and adjust their interpretation accordingly.

7) The current data interpretation relies heavily on the differences and changes in O:C (and H:C) values and the behaviour of f44 and f43. But there is no discussion of uncertainties/measurement errors for these values. Also, the authors do not state if they calculated the values for all data points, or did they apply a minimum mass concentration. Typically, O:C values become very noisy at AMS mass concentration < 0.5 ug m-3. I.e., the values at the start of each experiment may be less reliable. The authors need to add a comprehensive discussion about the uncertainties of O:C and H:C.

8) The authors completely ignore the changes in RH during their experiments. This may be fine for the constant temperature experiments as all are at RH < 15% (i.e., dry conditions). But for the ramp experiments this has to be considered as RH values as high as 80% (end of Exp 2.1) will lead to considerable amounts of water in the particles. The ramp experiments 2.1 - 2.3 are technically evaporation/condensation experiments assuming that most chemical processes have ceased that late in the experiment. Uptake and evaporation of vapours is strongly connected to the phase state of the particle (solid vs. liquid like) and will change with temperature and RH (Li et al., 2019; Zhao et al., 2019). Analysing the change in contribution of the factors should yield more insights into the effect of temperature and RH. The authors need to at least mention the possible impact of changes in RH during/between experiments.

9) It is good that the authors present results from other than the chosen PMF solution. But they do not provide enough information why the 4-factor solution was selected. Providing Scaled Residual values for all ions is a good start for such a discussion. However, I am not capable of deriving anything from the presented graphs (Fig S2-S4) other than that there are many red boxes and lines. As this is HR data, most boxes overlap making it very difficult to see any differences between the three figures. I recommend looking instead at the Relative and Scaled residual as time series. This will also show more directly for which experiment the additional factors improve the reconstruction. There needs to be more discussion (in the SI material) about why the 4-factor solution was chosen, and the presentation of the supporting graphics needs to be improved to be more accessible for the reader.

10) The Authors compare their data to ambient data by Lee et al. (2016) and conclude that their mass spectra are highly similar to those presented in that paper. This claim is difficult to judge as they are not providing the metric that they used for such comparisons. However, I agree that "by eye" the main patterns for Factor 1 and BSOA1 are similar even though the marker ion m/z 91 = $C_7H_7^+$ is missing from Factor 1. But Factor 3 exhibits distinct differences (44 higher than 43, higher 55, 27 and 29). And how is 0.39 & 1.59 (O:C and H:C of factor 3) "highly similar" to 0.56 & 1.56 (O:C and H:C of BSOA1)? These values correspond to OSC of -0.80 (factor 3) vs -0.44 (BSOA1). Comparing the van Krevelen diagrams shows that all measurement points in this study (Fig 4b) are outside of the range of measurements in Lee et al. (grey and orange dots in Fig 3). Only Factor 1 (the most oxidised here) is near the edge of the distribution and has similar O:H and H:C values as BSOA1 (the least oxidised in Lee et al, 2019). The authors have to find a better suited example to compare to their results.

11) The authors should compare their results with the findings of Zhao et al. (2019)

Specific comments:

1) What was the time resolution of the AMS?

2) Why was 35 min chosen to start the T ramp in Exp 1.4 and 1.5? How much of the precursor was still left at that time?

3) How large are the uncertainties of estimating density from O:C&H:C? (see also Major comment 7). Is this uncertainty smaller than what would be expected from using average RIE&CE values? Did you collect particle time of flight (pToF) data? Could you not compare the vacuum aerodynamic diameter with SMPS data? Without an uncertainty estimate such small changes as described in Fig S9 cannot be interpreted.

4) OSC can be calculated based on O:C and H:C. Are the trends in OSC during an experiment different from those in the density you calculate?

5) Your Q/Qexp values seem to level of at a value a lot larger than 1 which may be caused by using a too small measurement error value. Have you checked your measurement error estimate? Could there be additional sources of uncertainty?

6) In addition to Q/Qexp values the ratio of explained variance to total variance may be worth looking at when discussing which PMF solution was selected.

7) The mass spectra presented in e.g. Fig 1b or Fig 3 look like the HR data was recombined to UMR resolution and colour coded by family. If this is the case, this needs to be clearly stated.

8) Did you really use m/z 43 and m/z 44 derived from UMR analysis (using UMR frag tables) for the tracer ion analysis (page 8)? Or did you run the HR analysis and recombine to UMR signals? This needs to be stated.

9) For the second set of ramp experiments (Exp 1.5 and 1.6), it is of interest how much of the gas phase chemistry had already happened when you started the ramps as there should be some temperature dependence for the kinetics. Was all a-pinene consumed? Also, what were the starting gas phase compositions? Was the same amount/type of HOMs formed?

10) As the numbering of Factors in the PMF solutions does not have a meaning (to

my knowledge), you should consider a meaningful numbering, e.g. by the (assumed) volatility of the factors or the degree of oxidation (OSc). This will make the factor description and comparison easier to understand for the reader.

11) It is difficult to see how the relative contribution of the factors change with time in Fig 2. Add the "unstacked" factor time series to the SI material. This will help especially with interpreting the T ramp experiments.

12) What happened in Exp 3.3 around 50 min? There is a bump/shoulder in the time series.

13) When you are interpreting the simultaneous O:C and H:C change for the 50ppb experiments (page 7 last paragraph) you should take into account that in this specific case there is very little change in OSC, i.e., that the average degree of oxidation does not change. This may point towards non-oxidative reactions being the main source for the composition change. Or it might mean that at the same time more oxidised compounds (higher O:C than already in particles) are condensing as they are formed by ongoing oxidation reactions in the gas phase and also more volatile compounds (higher H:C than already in the particles) partition due to the increasing cOA. The net change could then be as you observe.

14) If you argue that compounds condensing at low temperatures are not well captured by the elemental analysis parameterisation, that would also be relevant for ambient measurements. 0 °C is not that uncommon during wintertime or nights, and the SOA mass loadings in your high concentration experiments are not that far off from ambient loadings in more polluted regions. Is it not more likely that you are simply observing condensation of more volatile and less oxidized species in these experiments? The 50 ppb and -15 °C case should have the least volatile compounds condensing as cOA was highest and C* values will be lowest. At higher temperatures, the effect vanishes as that compound class no longer partitions into the particle phase. You may just be probing a different range compounds at the lowest temperature (i.e., compounds that

are too volatile to condense at 20 °C).

15) Why were the T ramps during SOA formation done at 10 ppb and the T ramps at the end of the experiments at 50 ppb?

16) The slope of the O:C values for Exp 2.2 and 2.3 seems to change when T crosses 0 °C. For Exp 2.1 it is hard to say. Could there be an effect of "frozen" particles?

17) Figs 4b and 6b reveal some differences between "same" experiments which are not addressed in the text. Exp 2.1 and 3.1 show a bigger difference between each other than Exp 2.1 and 1.1 (Fig 5). In Fig 6, Exp 2.1 data is not visible. However, Exp 2.2 has decreasing f44 and f43, but Exp 3.2 is constant. f44 decreases in Exp 3.3 as much as it increases in Exp 2.2. I would not call that "good reproducibility" (page 8).

18) In Fig 6b, the arrow indicates a decrease of f43 for Exp 3.2 but the text states an increase for all experiments at -15 °C. Or is the arrow the wrong way round?

19) How much of the SOA mass is explained by the acids and esters detected in LC-MS? It is not simply (18+4)% and (38+11)%

Language and presentation:

+ Decide if you want to separate adverbial terms at the start of a sentence by a comma. In many cases, you used a comma. In other cases, you did not.

+ Reference entries to Quelever and Rosati are missing.

+ page 2 line 8 "and has it been" -> "and it has been"

+ page 2 line 10 OH and NO3 are not negative ions but radical. Use "·" (dot) not "-"

+ page 2 line 17f put the e.g part into commas or remove comma before "because"

+ page 4 line 35 "The number of factors are chosen. . ." -> is chosen

+ page 5 line 21&23, page 6 line 14 you use the term "CHO" in contrast to CHOgt1. But later you use "CHO1" several times. -> Choose one label.

+ page 6 lines 21-28 this should be together with the previous paragraph as it is still describing the factor MS. Do not just start a new paragraph become the current one is feeling too long.

+ page7 line 36/37 "hydration reactions with carbonyls" -> hydration reactions of carbonyls

+ page 8 line 30 "Alfarra 2004" -> Alfarra et al. 2004

+ page 8 line 40 "No clear tendencies" -> should be trends in this context

+ page 9 line 5 "(experiments 1.2, 2.2, and 2.3)" -> should be 3.2?

+ page 9 line18f "...to identify and quantify the 10 carboxylic acids which are regarded as some of the most important in $\alpha$-pinene derived SOA..." This sentence does not work. Most important what? Constituents?

+ Fig 4: Are the legend descriptions of Exp3.2&3.3 switched?

+ Fig 4: The black stroke around dark blue triangles is very hard to see -> use lighter color for -15 °C experiment (same issue in Fig 6).

+ Fig 4b: Do not put the yellow arrow on top of blue triangles

+ Fig 5: You use Kelvin as temperature unit while the text and labels use °C. -> Choose one unit.

+ Fig 5: The yellow dashed line is hidden by blue diamonds -> adjust y scale a bit to make it visible.

+ Fig 6b: where are the Exp3.1 data points? I only see red triangles with black border.

+ Fig 7 The term AMS m/z 44 *[factor] is a little bit confusing as the text speaks of (PMF) factors. maybe change this to [correction factor] or [scaling factor]

+ all SI Fig with PMF results: y axis is mass concentration not Mass. That is "lab-slang". You can easily avoid too long y axis label by introducing an abbreviation for

mass concentration (e.g. cm or corg).

+ all SI Fig with PMF results: experiment labels are 1-8 which is different from main text -> adjust labels.

References:

Bougiatioti, A., Stavroulas, I., Kostenidou, E., Zarmpas, P., Theodosi, C., Kouvarakis, G., Canonaco, F., Prévôt, A. S. H., Nenes, A., Pandis, S. N. and Mihalopoulos, N.: Processing of biomass-burning aerosol in the eastern Mediterranean during summertime, Atmos. Chem. Phys., 14(9), 4793–4807, doi:10.5194/acp-14-4793-2014, 2014. Bruns, E. A., El Haddad, I., Keller, A., Klein, F., Kumar, N. K., Pieber, S. M., Corbin, J. C., Slowik, J. G., Brune, W. H., Baltensperger, U. and Prévôt, A. S. H.: Inter-comparison of laboratory smog chamber and flow reactor systems on organic aerosol yield and composition, Atmos. Meas. Tech., 8(6), 2315–2332, doi:10.5194/amt-8-2315-2015, 2015. Kostenidou, E., Lee, B.-H., Engelhart, G. J., Pierce, J. R. and Pandis, S. N.: Mass Spectra Deconvolution of Low, Medium, and High Volatility Biogenic Secondary Organic Aerosol, Environ. Sci. Technol., 43(13), 4884–4889, doi:10.1021/es803676g, 2009.

Kristensen, K., Jensen, L. N., Glasius, M. and Bilde, M.: The effect of sub-zero temperature on the formation and composition of secondary organic aerosol from ozonolysis of alpha-pinene, Environ. Sci. Process. Impacts, 19(10), 1220–1234, doi:10.1039/c7em00231a, 2017. Kristensen, K., Jensen, L., Quéléver, L. L., Christiansen, S., Rosati, B., Elm, J., Teiwes, R., Pedersen, H., Glasius, M., Ehn, M. and Bilde, M.: The Aarhus Chamber Campaign on Highly Oxidized Multifunctional Organic Molecules and Aerosols (ACCHA): Particle Formation and Detailed Chemical Composition at Different Temperatures, Atmos. Chem. Phys. Discuss., 1–22, doi:10.5194/acp-2020-99, 2020.

Kroll, J. H., Donahue, N. M., Jimenez, J. L., Kessler, S. H., Canagaratna, M. R., Wilson, K. R., Altieri, K. E., Mazzoleni, L. R., Wozniak, A. S., Bluhm, H., Mysak, E. R.,

Smith, J. D., Kolb, C. E. and Worsnop, D. R.: Carbon oxidation state as a metric for describing the chemistry of atmospheric organic aerosol, Nat. Chem., 3(2), 133–139, doi:10.1038/nchem.948, 2011.

Lee, A. K. Y., Abbatt, J. P. D., Leaitch, W. R., Li, S. M., Sjostedt, S. J., Wentzell, J. J. B., Liggio, J. and Macdonald, A. M.: Substantial secondary organic aerosol formation in a coniferous forest: Observations of both day- and nighttime chemistry, Atmos. Chem. Phys., 16(11), 6721–6733, doi:10.5194/acp-16-6721-2016, 2016. Li, Z., Tikkanen, O. P., Buchholz, A., Hao, L., Kari, E., Yli-Juuti, T. and Virtanen, A.: Effect of Decreased Temperature on the Evaporation of $\alpha$-Pinene Secondary Organic Aerosol Particles, ACS Earth Sp. Chem., 3(12), 2775–2785, doi:10.1021/acsearthspacechem.9b00240, 2019.

Quelever, L. L. J., Kristensen, K., Normann Jensen, L., Rosati, B., Teiwes, R., Daellenbach, K. R., Peräkylä, O., Roldin, P., Bossi, R., Pedersen, H. B., Glasius, M., Bilde, M. and Ehn, M.: Effect of temperature on the formation of highly oxygenated organic molecules (HOMs) from alpha-pinene ozonolysis, Atmos. Chem. Phys., 19(11), 7609–7625, doi:10.5194/acp-19-7609-2019, 2019.

Wan, K. X., Vidavsky, I. and Gross, M. L.: Comparing similar spectra: From similarity index to spectral contrast angle, J. Am. Soc. Mass Spectrom., 13(1), 85–88, doi:10.1016/S1044-0305(01)00327-0, 2002.

Zhao, Z., Le, C., Xu, Q., Peng, W., Jiang, H., Lin, Y.-H., Cocker, D. R. and Zhang, H.: Compositional Evolution of Secondary Organic Aerosol as Temperature and Relative Humidity Cycle in Atmospherically Relevant Ranges, ACS Earth Sp. Chem., 3(11), 2549–2558, doi:10.1021/acsearthspacechem.9b00232, 2019.

---

## Author Comment (AC1) · 20 Sep 2020

Jensen et al.: Temperature and VOC concentrations as controlling factors for chemical composition of alpha-pinene derived secondary organic aerosol

**Reply to review by Anonymous Referee #1**

*We thank the reviewer for the constructive comments, which we have addressed in a point-by-point fashion below (responses written in italics). We have modified the manuscript accordingly.*

**Summary**
This paper investigates the effects of temperature and alpha-pinene concentration on the chemical composition of SOA particles formed via dark ozonolysis in a series of chamber studies. Experiments were performed in the dark at two concentrations, 10 ppb and 50 ppb, in the presence of ozone. Three constant temperatures were analyzed (20, 0, and -15 C), and several temperature ramps were induced at various points during the ozonolysis. Chemical composition was investigated in real-time using an HR-ToF-AMS, and offline chemical composition of filter samples was performed via LC-MS. The authors apply PMF to the AMS data in order to determine controlling factors behind the chemical composition. They discuss a four-factor solution, where two of the factors correspond to temperature and two factors correspond to alpha-pinene concentration. The authors argue that the temperature at which particles initially form is more important for determining particle composition than changes in temperature after the initial formation. However, they do find that temperature ramps lead to slight changes in oxidation levels.
The authors also compare AMS derived estimates of organic acid content to off-line LC-MS analysis of filters and find good agreement. They conclude that high temperatures and low initial alpha-pinene concentrations are associated with more oxidized SOA.

**General Comments**
This paper is not suitable for publication in its present form. In general, a more rigorous and thorough discussion of results with consideration of context would be helpful. There is a tendency in this manuscript to state results and not interpret them.
I also take issue with some of the findings. The second to last sentence of this manuscript is "This work confirms that the particle chemical composition is dependent on precursor concentration and particle mass loading." This is not explored in the manuscript.

*Based on the reviewer's comments we have modified the manuscript in several places with focus on a more clear and thorough discussion of the results. Moreover, we would like to specify that a deeper discussion of some of the results can be found in the companion paper by Kristensen at al. 2020.*

*Regarding the second last sentence we agree that this is not explored in the manuscript - what we wanted to write is precursor concentration and temperature. This is now corrected in the manuscript: "The companion paper also shows and discusses how increased VOC precursor concentration leads to a higher SOA particle mass loading consistent with findings in previous chamber studies (Kristensen et al. 2020).*

The authors state that "Factor 1" is a temperature factor, however this does not seem to be the case. Comparing experiment 1.4 to 3.1 (same constant temperature) Factor1 appears to change more than a factor of 10 for a factor of 5 change in precursor VOC concentration.

Further, Factor 1 does not change during the lower VOC (10ppb) temperature ramp experiments. Factor 1 is essentially the same for experiments 1.4 vs.1.5, though this is a little hard to discern because the y-axes are different.

*Thanks for pointing out that it was not clear in the text why we refer to Factor 1 as a temperature factor. It is important to notice, that we do not compare absolute contributions of the factors but rather their relative contributions to the total mass. For comparison of experiments 1.4 and 1.5 it is the relative contribution of Factor 1 to total mass that should be compared. We see a relatively higher contribution from the high temperature factor (1) in experiment 1.4 and a relatively higher contribution from the low temperature factor (4) in experiment 1.5. Additionally, by comparing the warm experiments (1.1 and 3.1, both 20 °C but different VOC concentration) it shows that Factor 1 contributes significantly to the total mass in both experiments - and in the corresponding cold experiments (1.3 and 3.3, both 20 °C but different VOC concentration) Factor 1 is almost non-existing. We have modified the text to make this more clear, e.g. we have made the following paragraph more clear: "According to their appearance and relative contribution to total SOA mass Factors 1 and 4 will in the following discussion be referred to as "temperature factors" and Factors 2 and 3 will be referred to as "concentration factors". For example, according to Figures 2 and 3, at both α-pinene concentrations Factor 1 makes up a significant fraction of the particle mass in the 20 °C experiments but plays a minor role in the colder experiments and therefore Factor 1 will be referred to as "high temperature factor" in this manuscript. The significant contribution of Factor 1 to the SOA mass at high temperature is in agreement with the fact that this factor is mostly dominated by ions from oxidized species (i.e. high intensity of CHO and CHOgt1 ion groups at m/z 28, 29, 43, 44, 55, and 83). Among all factors, Factor 1 has the highest O:C ratio (0.56), OSC (-0.53), f43 (14 %), and f44 (9 %) and therefore the chemical species represented by Factor 1 (i.e. related to high temperature) are likely the most oxidized and least volatile (c\*) entities present in the SOA."*

The authors admit that a five- or six-factor solution provides a better explanation for the data than the four-factor solution presented in the manuscript, but do not adequately explore why the higher-order factor solutions are discarded.

*The larger number of factors are not selected because they do not provide any more interpretable information about the composition. This is now explained more detailed in the Supplementary Information of the revised manuscript.*

Ramp start-to-end RH is presented in Table 1, but it is not explored in the manuscript. RH values change drastically during the temperature ramp, but this information is not used in the analysis. Is it possible that the ignored factors from the PMF solution are instead factors relating to RH? If so, they should be included in the analysis and their importance should be discussed, or a more adequate reason for discarding them should be addressed.

*In the PMF analysis we include only temperature ramps performed early in the experiments (i.e. 1.4 and 1.5): the largest variation in relative humidity is observed in experiment 1.4 where relative humidity changes from ~1 to ~30 % during the ramp (cooling). That is, all experiments included in the PMF analysis is conducted at low (<30 %) relative humidity. If a factor was related to relative humidity we would expect this factor to appear during temperature ramping to low temperature (experiment 1.4) and with time in the high concentration experiment as the RH increases slightly during these experiment. We do not observe the appearance of such a factor in neither the four, five, or six factor solutions.*

Atmospheric Chemistry and Physics is an applied journal and it surprises me the authors provide no atmospherically relevant context for the selected temperatures, initial hydrocarbon concentrations, etc.

*We agree with the reviewer that such information is lacking. We have added the following text at the end of the introduction: "The ACCHA campaign focuses on temperatures from 20°C to - 15°C representing conditions relevant in the boreal forest regions (Portillo-Estrada et al. 2013) and at atmospheric relevant concentrations of α-pinene ranging from 10 ppb, representing the higher concentrations in boreal forest (Kourtchev et al. 2016), to 50 ppb.*

The authors state early in their manuscript that oxidation levels of SOA from a-pinene ozonlysis decrease with increased particle mass loadings. The authors' findings do not seem to be consistent with this. Why?

*We assume the reviewer refer to our discussion of the absortive partitioning framework which shows that higher particle mass loading can serve as a driver of condensation of more volatile less oxidized species to the condensed phase. According to the Clausius Clapeyron equation vapor pressure is expected to decrease with decreasing temperature and would thus be expected to lead to condensation of less oxidized compounds to the condensed phase. This is actually, what we observe. To make this more clear we have revised the text to read: "In accordance with expectations based on the absortive equilibrium framework (1), previous work (Shilling et al. 2009), and observations from the PMF analysis SOA formed from low α-pinene concentration (10 ppb) and higher temperature are associated with higher O:C ratios compared to SOA formed from the high α-pinene concentration (50 ppb) at lower temperatures. The plot shows that the SOA consists of less oxidized compounds in experiments conducted at low temperature and high α-pinene concentration, compared to high temperature and low α-pinene concentration."*

**Specific comments and line numbers follow:**
Page 2, Line 6: SOA is also formed through the uptake of water-soluble organic gases into condensed-phase water.

*Based on this comment from the reviewer we have modified the sentence and it now reads: "Due to their low vapor pressures, some of the gas phase oxidation products may partition onto already existing particles by condensation or reactive uptake and contribute to particle growth." (Hallquist et al. 2009)*

Page 2, Line 7: The 1995 Guenther manuscript describing a biogenic emissions model is not the right reference for "SOA forms from biogenic and anthropogenic precursors".

*We agree, and we have replaced this reference with a more general reference namely the text book by Seinfeld and Pandis.*

Page 2, Line 10: Hydroxyl radical (OH) should have a dot, not a minus sign.

*Thanks, corrected.*

Page 3, Line 28-32: Why were different precursor concentrations chosen for the temperature-ramp experiments? If precursor concentration is playing an important role in chemical

composition, then why aren't all the temperature-ramp experiments per-formed using the same initial alpha-pinene concentration? Here, some are done with10 ppb and some with 50 ppb. Why are these conditions relevant?

*In the revised manuscript we have only included the two 10 ppb a-pinene temperature ramps (1.4 and 1.5) initiated around 35 minutes after starting the experiment (see also reply to reviewer 2). As now stated in the introduction, a concentration of 10 ppb corresponds to the level which can appear in boreal forests.*

Page 4, Line 36 – Page 5, Line 1: According to Table 1, these experiments also saw a wide range of RH, which could lead to the condensation of water onto the particles and subsequent aqueous-phase reactions. Is it possible that Factors 5 and 6 are related to RH? If so, please address this in the manuscript, or provide a justification for why these factors and the large range of RH values can be ignored.

*The ramps with a large change in relative humidity have been omitted from the manuscript (see also answer to reviewer 2). In all experiments included in the PMF analysis RH < 30 %.*

Page 5, starting at Line 14: The authors state, "Under all conditions (i.e. in all experiments), the AMS derived SOA densities are in the range of 1.1 to 1.3 g cm-3. Figure 1 indicates that SOA density is < 1.1 for the first 0.5 hour.

*The sentence has been modified to:*
*"The AMS derived SOA densities at mass peak are of the order of 1.1- 1.3 g cm$^{-3}$. Figure S9 suggests a slight increase in density with higher experimental temperature as well as a slightly higher density for the particles formed at low α-pinene concentration (10 ppb) compared to high α-pinene concentration (50 ppb)."*
*See also reply to comment below in relation to SOA density.*

Page 5, Line 32-35: Please provide a reasoning for using only some of the temperature ramp experiments in the PMF analysis.
*We have re-considered the ramp experiments conducted at 50 ppb α-pinene and decided to omit these three ramps from the analysis. The main reason is that these ramps were performed after filter sampling for the off-line analysis. Filter sampling results in significant reduction in the chamber volume in a short time, which means that conditions in the chamber are different after filter sampling.*

*PMF analysis was conducted on experimental data during the first up to ~250 minutes of the experiments. Therefore, the two 10 ppb α-pinene ramp experiments, where the temperature ramp was started ~35 minutes after SOA formation and completed after ~160 min, are included in the PMF analysis.*

Page 6, Line 29-30: What contributions from f44 and m/z ratios do Factors 5 and 6have?

*As now explained in the Supplementary Information the 5- and 6-factor solutions do not provide additional information and we therefore focus the analysis on f44 and m/z ratios from the 4-factor solution only.*

Page 7, Line 12-14: Please provide a discussion regarding why it makes sense to have less-oxidized SOA at low temperatures and high alpha-pinene concentrations. It is interesting that

this is the case, but a scientific justification is missing and should be discussed. Is this a matter of kinetics and competition for oxidants?

*See answer to comments above and relation to partitioning considerations.*

Page 7, Line 31: Why is no OH scavenger used, and how does potentially changing [OH] confound interpretation of O:C ratios and factors? The authors provide some speculation on Page 7, but the context is insufficient.

*An OH-scavenger can be added to chamber experiments to study ozonolysis reactions in isolation while experiments without scavenger are generally considered more realistic although also more complex. The ACCHA campaign was conducted without use of OH-scavenger so that both ozone and OH-radicals acted as oxidizing agents during the course of the experiments. The ratio between oxidation by OH- radicals and $O_3$ was modelled in selected experiments using the master chemical mechanism v3.3.1.It was found that the ratio of VOC oxidized by $O_3$ relative to that oxidized by OH-radicals was ~2:1 independent of temperature and precursor concentration(Quéléver et al. 2019). This is now mentioned explicitly in the text:*

*"Modelling suggests, however, that the OH oxidation is not more pronounced at low temperature (0 °C) compared to high temperature (20 °C) (Quéléver et al., 2019),which makes this a less likely explanation for the continuous increase in the O:C ratio and H:C ratio in the cold experiments. More specifically, the ratio of VOC oxidized by O3 relative to that oxidized by OH-radicals was ~2:1 independent of precursor concentration and temperature."*

Page 8, Line 18-24: A lot of time and energy is spent on discussing the reasons behind changes in the O:C and H:C ratios during temperature ramps in previous paragraphs. However, here it is stated that the changes are small and not very important. If that is the case, then why discuss the changes to the O:C and H:C ratios so extensively? It seems this section could be more focused.

*There are some changes in O:C and H:C ratios during the temperature ramping and we find it justified to discuss it. At the same time, it is also true that the changes are small compared to the differences between particles formed at different temperatures. To make this more clear we have modified the text in this section.*

Page 8, Line 39 – Page 9, Line 9: Please provide a reasoning for why a lower precursor concentration leads to more highly oxidized SOA. Is this because there is less competition for oxidants?

*Lower precursor concentration means less particle mass and thus it is only the more oxidized species that have low enough vapor pressure to condense, see discussion on partitioning framework above. We have added a reasoning in the text as requested by the reviewer (only bold text):*

*"This suggests that acid-derived functionalities are more prevalent in α- pinene SOA formed at low precursor concentration (and thus low particle mass loading) which is consistent with less partitioning of the more volatile, less oxidized compounds to the particle phase."*

Page 10, Line 9-12: The authors state that organic acids with heavier backbones are formed at lower temperatures. Why does this make sense? Is this due to volatility reasons?

*The finding is consistent with the lower O:C ratio at lower temperature as reported by Kristensen et al. 2017 in the AURA chamber and recently also reported in studies from the CLOUD chamber at CERN (Simon et al. ACP 2020). This trend is attributed to the effect of temperature on peroxy radical chemistry.*

Figures and Tables: Table 1: Do I understand correctly that no experiment was repeated and there is no demonstrated repeatability?

*The ACCHA campaign was carefully designed with repeatability checks in mind. For reproducibility with respect to time evolution of SOA mass, α-pinene concentration, $O_3$ concentration, temperature and relative humidity see supplementary material in Kristensen et al. 2020.*

*The following text has been added to the manuscript:*
*"For reproducibility with respect to formation of SOA mass as well as loss rates of a-pinene and $O_3$ see Kristensen et al. (2020)."*

Figure 1: Particle density is presented with four significant digits and no estimate of uncertainty. There should be uncertainty bars or some other representation.

*We agree that four significant digits is too much. We have removed the density in Figure 1a.*

Figure 4: In panel b, what is the black line? It is not labeled.

*The black line is the same as in panel b, but it should be noted that the scales on the axes are different. To make this more clear, the caption has been modified adding:*

*"Notice the different scales of the axes in panels a and b."*

Figure 6: In panel b, the black line is not defined. Why is the black line in panel b different from panel a, if they are both based on Heald and Ng?

*The black lines are the same - it is the scales on the axes that are different in panels a and b. To make this more clear, the caption has been modified adding:*

*"Notice the different scales of the axes in panels a and b."*

In panel a, it is unclear how this figure supports the author's conclusions regarding the oxidation level of SOA in each experiment.

*Panel a) is included to show where the results from this work fall in relation to the lines suggested by Heald et al. (2010) and Ng et al. (2011). If the axes in panel b has been changed to allow for inclusion of both of these lines the detailed information about time evolution would be lost.*

Supplemental Information: Figures S5 and S7. Could these extra factors be attributable to RH? They seem to play an consequential role in mass loading

*This question has already been answered in a former question.*

References

Kristensen, K.; Jensen, L. N.; Quéléver, L. L. J.; Christiansen, S.; Rosati, B.; Elm, J.; Teiwes, R.; Pedersen, H. B.; Glasius, M.; Ehn, M.; Bilde, M. 2020. The Aarhus Chamber Campaign on Highly Oxidized Multifunctional Organic Molecules and Aerosols (ACCHA): Particle Formation and Detailed Chemical Composition at Different Temperatures. *Atmospheric Chemistry and Physics* Accepted.

Kourtchev, I.; Giorio, C.; Manninen, A.; Wilson, E.; Mahon, B.; Aalto, J.; Kajos, M.; Venables, D.; Ruuskanen, T.; Levula, J.; Loponen, M.; Connors, S.; Harris, N.; Zhao, D.; Kiendler-Scharr, A.; Mentel, T.; Rudich, Y.; Hallquist, M.; Doussin, J.-F.; Maenhaut, W.; Bäck, J.; Petäjä, T.; Wenger, J.; Kulmala, M.; Kalberer, M. 2016. Enhanced Volatile Organic Compounds emissions and organic aerosol mass increase the oligomer content of atmospheric aerosols. *Scientific Reports 6* (1), 35038.*.*

Portillo-Estrada, M.; Korhonen, J. F. J.; Pihlatie, M.; Pumpanen, J.; Frumau, A. K. F.; Morillas, L.; Tosens, T.; Niinemets, Ü. 2013. Inter- and intra-annual variations in canopy fine litterfall and carbon and nitrogen inputs to the forest floor in two European coniferous forests. *Annals of Forest Science 70* (4), 367.

Quelever, L. L. J.; Kristensen, K.; Jensen, L. N.; Rosati, B.; Teiwes, R.; Daellenbach, K. R.; Perakyla, O.; Roldin, P.; Bossi, R.; Pedersen, H. B.; Glasius, M.; Bilde, M.; Ehn, M. 2019. Effect of temperature on the formation of highly oxygenated organic molecules (HOMs) from alpha-pinene ozonolysis. *Atmospheric Chemistry and Physics 19* (11), 7609.

Seinfeld, J.H., Pandis S.N. 2016. Atmospheric chemistry and physics : from air pollution to climate change; Third edition ed.; Wiley.

Simon, M.; Dada, L.; Heinritzi, M.; Scholz, W.; Stolzenburg, D.; Fischer, L.; Wagner, A. C.; Kürten, A.; Rörup, B.; He, X. C.; Almeida, J.; Baalbaki, R.; Baccarini, A.; Bauer, P. S.; Beck, L.; Bergen, A.; Bianchi, F.; Bräkling, S.; Brilke, S.; Caudillo, L.; Chen, D.; Chu, B.; Dias, A.; Draper, D. C.; Duplissy, J.; El-Haddad, I.; Finkenzeller, H.; Frege, C.; Gonzalez-Carracedo, L.; Gordon, H.; Granzin, M.; Hakala, J.; Hofbauer, V.; Hoyle, C. R.; Kim, C.; Kong, W.; Lamkaddam, H.; Lee, C. P.; Lehtipalo, K.; Leiminger, M.; Mai, H.; Manninen, H. E.; Marie, G.; Marten, R.; Mentler, B.; Molteni, U.; Nichman, L.; Nie, W.; Ojdanic, A.; Onnela, A.; Partoll, E.; Petäjä, T.; Pfeifer, J.; Philippov, M.; Quéléver, L. L. J.; Ranjithkumar, A.; Rissanen, M. P.; Schallhart, S.; Schobesberger, S.; Schuchmann, S.; Shen, J.; Sipilä, M.; Steiner, G.; Stozhkov, Y.; Tauber, C.; Tham, Y. J.; Tomé, A. R.; Vazquez-Pufleau, M.; Vogel, A. L.; Wagner, R.; Wang, M.; Wang, D. S.; Wang, Y.; Weber, S. K.; Wu, Y.; Xiao, M.; Yan, C.; Ye, P.; Ye, Q.; Zauner-Wieczorek, M.; Zhou, X.; Baltensperger, U.; Dommen, J.; Flagan, R. C.; Hansel, A.; Kulmala, M.; Volkamer, R.; Winkler, P. M.; Worsnop, D. R.; Donahue, N. M.; Kirkby, J.; Curtius, J. 2020. Molecular understanding of new-particle formation from α-pinene between −50 and +25°C. *Atmospheric Chemistry andPhysiscs. 20* (15), 9183.

---

## Author Comment (AC2) · 20 Sep 2020

**Summary:** The authors present a very interesting dataset of secondary organic aerosol (SOA) composition measurements with a High-resolution Time of flight AMS (short: AMS) and offline filters. They investigated the evolution of a-pinene SOA generated via dark ozonolysis in a temperature-controlled chamber either at constant temperatures (20◦C, 0◦C, and -15◦C) or with controlled temperature ramps. They compared the endpoint composition with offline filter measurements and apply Positive Matrix Factorisation (PMF) analysis to the AMS data. They find PMF factors describing the difference observed in composition due to the changing temperature and factors that capture the impact of different precursor concentrations. They conclude from the temperature ramp experiments that the temperature at which the particle formation is more important for the overall particle composition than the final temperature. I think this is a great data set with the potential to give important insights into how temperature and precursor concentration control the chemical composition of a-pinene SOA particles which is important to understand when comparing chamber experiments to ambient measurements. Unfortunately, the presented analysis and interpretation of this data set are somewhat superficial. The authors mostly just describe the data but leave the reader wondering about the implications. I therefore request major revisions of the data interpretation and discussion part in addition to addressing my comments below before this paper can be published in ACP.

*We thank the reviewer for the constructive comments. Based on the comments of both reviewers we have significantly revised the manuscript for clarity. We have addressed the reviewers comments in a point by point fashion below (responses in italics).*

**Major Comments:**

1) One reason the discussion of the results is so unsatisfactory is that the authors do not use the additional information available for these experiments. As one example, take the only small changes in O:C observed in the temperature ramp experiments 2.1- 2.3. How much did the SOA mass concentration / particle size change in that time? I.e., how much was the existing SOA composition perturbed by the T ramp? With that in mind, how much change can be expected in a value like O:C? I.e., if cooling from20◦C to -15◦C leads to the condensation of x% of compounds with lower O:C how much does that change the O:C of the original particles? Is O:C sensitive enough to monitor such changes? This is just one example of how the authors can improve the interpretation and explanation of their results. I request that the authors utilise the other available information (part of it is already published in Kristensen et al. (2020) and Quelever et al. (2019)) to support their findings and strengthen the discussion.

*We have significantly revised the manuscript to make it clearer. We have reconsidered the ramps in experiments 2.1-2.3 and decided to leave them out. The main reason is that these ramps were performed late in the experiments (>200 min) after filter sampling for the off-line analysis. Filter sampling results in significant reduction in the chamber volume in a short time, which means that conditions in the chamber are different after filter sampling and not comparable to the first 200 minutes of the experiments.*
*For Exp. 1.4 and 1.5 SOA formation is still ongoing during the ramp and therefore such mass analysis requires more advanced modelling which is outside the scope of the current study. Omitting the temperature ramps in Exp. 2.1-2.3 do not change the outcome or conclusions of the paper – in fact we believe it improves focus and readability of the paper.*

2) A second reason is that the authors do not use the results of their presented dataset to the full extent possible. They conducted PMF analysis and concluded that the grouping of the factors is connected to

the SOA formation temperature. Why do they not use this to investigate composition changes in the ramp experiments (Exp 1.5&1.6,C2 2.1-2.3)?

*As explained above the temperature ramps in Exp. 2.1-2.3 are omitted from the manuscript. Composition changes during the ramp experiments 1.4-1.5 are included in the PMF analysis as stated in section 3.1.*

The PMF chapter (3.1) is mostly describing the marker ions in the different factors and comparing the factors with each other (and one ambient example). But there is no analysis of the temporal evolution of the contribution of the factors during the experiments. Chapter 3.2 is instead focusing on the changes in O:C and the two tracer ions m/z 43 and m/z 44. Kristensen et al. (2017) already covers the temperature dependence of the acid and ester dimer formation in detail. The fact that the AMS f44values agrees with the LC-MS results is nice. I request a more thorough interpretation of the changes in PMF factor contributions during the experiments which will hopefully reveal the "bigger" picture of the composition changes with temperature and precursor concentration as the PMF analysis utilises all ions detected in the AMS and not just m/z 43 and m/z 44.

*We thank the reviewer for this comment. While the focus of the analysis was to understand the difference in composition between different experiments, we have now added more information about the insight learned from the PMF factor contributions within each experiment. The paragraph below has been added to the manuscript as follows:*

*"While not entirely decoupled, overall, the four PMF factors represent different main characteristics of the particle chemical composition associated with temperature and precursor concentration and provide a useful framework for discussing effects of temperature and VOC loading on SOA formation and properties in chamber experiments. Figure 2 shows that within each experiment, the relative contribution of the factors changes with time. While these relative ratios likely reflect the changes in SOA composition from nucleation (beginning of experiment), condensational growth (increase in mass concentration), and wall loss (decrease in mass concentration at end), changes due to ongoing gas phase chemistry may also affect observed trends in composition (Kristensen et al., 2020). In each chamber experiment, the correlation between the relative contribution of each factor and the aerosol mass concentration can be utilized to infer information about the relative volatilities of the species in each factor. For example, figure 2 shows that for experiments 1.1 and 1.2, the relative mass concentration ratio of Factor 1 to Factor 2 is largest at lower SOA mass concentrations. This suggests that the volatility of species related to Factor 1 is lower than Factor 2 species. Similarly, Figure 2 shows that for experiments 3.2 and 3.3 the relative mass concentration ratio of Factor 3 to Factor 4 is largest at lower SOA mass concentrations. This indicates that the volatility of Factor 3 species is lower than Factor 4 species. The relative volatilities of Factor 2 and Factor 3 species can be assessed by comparing the high precursor and aerosol mass concentration experiments (i.e. experiments 3.1) and the low precursor and aerosol mass concentration experiments. The fact that Factor 2 is dominant under the low concentration conditions indicates that species associated with Factor 2 have lower volatilities than those associated with Factor 3. Taken together these results suggest that the volatility (c\*) of the four factors increases in the following way: Factor 1 < Factor 2 < Factor 3 < Factor 4. The factors related to temperature variation (Factors 1 and Factor 4) show a larger difference in oxidation level than the factors related to α-pinene concentration, i.e. particle mass loading (Factors 2 and 3). This suggests that within the investigated conditions, differences in temperature (20 °C to -15 °C) have a larger effect on particle chemical composition than VOC concentration (10 and 50 ppb α-pinene).*

3) The paper presents many (factor) mass spectra and compares them stating that they are similar (or not). But the authors do not explain what metric they use to determine similarity (statistical method, ratios of marker ions, "by eye"). If they have not used a objective (mathematical) metric, I

recommend using the spectral contrast angle, theta,which is derived from the dot product of two mass spectra, A and B (Wan et al., 2002).A and B are considered similar if theta is between 0∘and 15∘, somewhat similar butwith important differences if theta is between 15∘ and 30∘, and different with thetavalues >30∘(Bougiatioti et al., 2014). This parameter (sometimes as cos theta with values between 0 and 1) has been used for AMS data before (e.g. Bruns et al., 2015;Kostenidou et al., 2009). Such a pairwise comparisons will also help with claims like that the composition of the 0∘C and -15∘C experiments are closer to each other than to the 20∘C case. I request that the authors specify how they determined similarity of mass spectra and provide values/graphs that show the degrees of similarity (can be inSI material).

*We have now added a comparison of the similarity of the mass spectra of the four factors based on the methods described on Wan et al. (2002) and Ulbrich et al. (2009). The results are presented in the supplementary and in the text the following paragraph is added:*

*"On a scale from 0 to 1, with 1 indicating highest similarity, the similarity is calculated to be between 0.86 and 0.97 across all factors in the m/z range from 12 to 115, using the method described in Ulbrich et al. (2009). By focusing only on the m/z > 44 similarities between 0.85 and 0.97 are obtained. In the Wan et al. (2002) calculation higher numbers indicates higher dissimilarity. Taken together, both methods suggests that the mass spectra of Factors 3 and 4 are the most similar and Factors 1 and 2 are the least similar."*

4) The explanation and labelling of the PMF factors as high/low concentration or temperature is a bit misleading and easily misunderstood. Their labelling/description makes it sound as if the effect of higher precursor concentration is covered by switching from Factor 3 to Factor 2. But the higher cOA (organic aerosol particle mass) also impacts the contribution of the other factors. As an example, Factor 4 (low T) increases its contribution at all temperatures when the VOC concentration is increased. In a similar way, the contribution of Factor 2 (high concentration) strongly depends on temperature (e.g. increasing for 10ppb cases with decreasing temperature). They authors do explain the connection of partitioning with C* and cOA in the introduction in general, but they do not clearly state how the factors fit into this framework. The authors need to clarify how and why the factors are related to cOA and temperature, i.e., interpret the factors with regard of volatility.

*We have revised the explanation of the naming of the factors to be clearer and have also connected the factors to their respective C* values as described in the answer to comment #2 above. It is also now explicitly stated that:*

*"As shown in Figure 2, the SOA observed under each experimental condition does not correspond to a single PMF factor. The SOA obtained under different experimental conditions are represented by a linear combination of multiple factors instead. The factors are clearly distinguished from each other, however, by consistent trends in their relative mass contributions to the SOA observed under the different experimental conditions. These trends are used in the interpretation and naming of the factors. According to their appearance and relative contribution to total SOA mass Factors 1 and 4 will in the following discussion be referred to as "temperature factors" and Factors 2 and 3 will be referred to as "concentration factors". "*

5) Continuing the point from the previous comment, from the limited explanations provided in the text, I derive that the authors made the assumption that the factorization in PMF depends on the condensation behaviour of the compounds which is governed by the temperature and the increasing cOA. I.e., the increase in contribution of factor4 with time in Exp 3.2 and 3.3 would thus be explained by higher volatility material condensing as cOA increases. But why would then Factor 2 decrease already before the peak in mass concentration is reached? Could that not be connected to ongoing chemical processes? In this paper, there is no information about the consumption of the precursor and thus on how long reactions may be ongoing. How do the different reaction rates at low/high temperature and/or different precursor concentrations affect the oxidation products? This is especially interesting for the T ramp experiments (1.5& 1.6). The authors have to at least mention if or how

ongoing chemistry would impact the PMF factors and volatility of reaction products. If the authors merely want to com-pare the final composition of the formed particles, they need to clarify that and provide information how they determine the "end point".

*We interpret the observed behavior as largely affected by a competition between condensation of high volatile material (Factor 4) as SOA mass increases (time evolution within one experiment) and as temperature is lowered (comparing 3.1, 3.2 and 3.3) with formation of new condensable species characteristic for high a-pinene concentration (Factor 3). The loss of a-pinene with time is shown in the supplementary material in Kristensen et al. (2020) and shows that there is still some a-pinene in the chamber after100 min when factor 3 peaks in Exp. 3.3.*
*The PMF factor analysis thus encompasses on-going gas phase chemistry in the chamber. This is more clearly stated in new text that has been added to the manuscript as follows:*

*"Figure 2 shows that within each experiment, the relative contribution of the factors changes with time. While these relative ratios likely reflect the changes in SOA composition from nucleation (beginning of experiment), condensational growth (increase in mass concentration), and wall loss (decrease in mass concentration at end), changes due to ongoing gas phase chemistry may also affect observed trends in composition (Kristensen et al., 2020)."*

6) As in many papers with AMS data, the authors use the O:C values as a parameter for the oxidation level of the investigated SOA particles. However, if the H:C values also change (as they point out when discussing the van Krevelen diagram in Fig 4b),the case is not that simple and the average oxidation state of carbon (OSC, Kroll et al. (2011)) may be the better parameter representing the degree of oxidation in the particles. As an example, for Exp 3.3 (50 ppb, -15 C) in Fig 4b, I read a change for O:C & H:C values from 0.28 & 1.62 to 0.33 & 1.705, respectively. Calculating OSC as $OSC = 2*O:C - H:C$ gives 1.05 and -1.06 for the start and end of the experiment, i.e.,the average oxidation state of carbon did not change. As another example, the OSC values of factors 2 and 4 are almost identical although the O:C values differ. Do the authors have a reason why O:C would be a better measure of the level of oxidation in their experiments than OSC? If not, they should change to using OSC to account for changes both in O:C and H:C and adjust their interpretation accordingly.

*This is a very good point. Thank you for the comment. The OSC has now been added in various places in the manuscript. In particular we have included a figure (5c) showing the evolution in OSC in the experiments.*

7) The current data interpretation relies heavily on the differences and changes in O:C (and H:C) values and the behaviour of f44 and f43. But there is no discussion of uncertainties/measurement errors for these values. Also, the authors do not state if they calculated the values for all data points, or did they apply a minimum mass concentration. Typically, O:C values become very noisy at AMS mass concentration <0.5 ug m-3. I.e., the values at the start of each experiment may be less reliable. The authors need to add a comprehensive discussion about the uncertainties of O:C and H:C.

*After the first 2-3 data points shown in the figures the standard deviation decreases to between ~0.5 and 3.0 %. Figures showing the evolution in the standard deviation are now added to the Supplementary Information.*

8) The authors completely ignore the changes in RH during their experiments. This may be fine for the constant temperature experiments as all are at RH < 15% (i.e., dry conditions). But for the ramp experiments this has to be considered as RH values as high as 80% (end of Exp 2.1) will lead to considerable amounts of water in the particles. The ramp experiments 2.1 - 2.3 are technically evaporation/condensation experiments assuming that most chemical processes have ceased that late in the experiment. Up-take and evaporation of vapours is strongly connected to the phase state of the particle (solid vs. liquid like) and will change with temperature and RH (Li et al., 2019; Zhao et al., 2019). Analysing the change in contribution of the factors should yield more insights into the effect of

temperature and RH. The authors need to at least mention the possible impact of changes in RH during/between experiments.

*As explained above we have decided to omit the temperature ramps in exp. 2.1-2.3 from the manuscript as they were performed after filter sampling. It means that the RH does not exceed 30 % in any of the experiments included in the manuscript.*

9) It is good that the authors present results from other than the chosen PMF solution. But they do not provide enough information why the 4-factor solution was selected. Providing Scaled Residual values for all ions is a good start for such a discussion. However, I am not capable of deriving anything from the presented graphs (Fig S2-S4) other than that there are many red boxes and lines. As this is HR data, most boxes overlap making it very difficult to see any differences between the three figures. I recommend looking instead at the Relative and Scaled residual as time series. This will also show more directly for which experiment the additional factors improve there construction. There needs to be more discussion (in the SI material) about why the 4-factor solution was chosen, and the presentation of the supporting graphics needs to be improved to be more accessible for the reader.

*The effect of number of factors on scaled residual as a function of time and m/z is now shown in the supplementary information.*

10) The Authors compare their data to ambient data by Lee et al. (2016) and conclude that their mass spectra are highly similar to those presented in that paper. This claim is difficult to judge as they are not providing the metric that they used for such comparisons. However, I agree that "by eye" the main patterns for Factor 1 and BSOA1 are similar even though the marker ion m/z 91 = C7H7+ is missing from Factor 1. But Factor 3 exhibits distinct differences (44 higher than 43, higher 55, 27 and 29). And howis 0.39 & 1.59 (O:C and H:C of factor 3) "highly similar" to 0.56 & 1.56 (O:C and H:Cof BSOA1)? These values correspond to OSC of -0.80 (factor 3) vs -0.44 (BSOA1).Comparing the van Krevelen diagrams shows that all measurement points in this study (Fig 4b) are outside of the range of measurements in Lee et al. (grey and orange dots in Fig 3). Only Factor 1 (the most oxidised here) is near the edge of the distribution and has similar O:H and H:C values as BSOA1 (the least oxidised in Lee et al, 2019).

The authors have to find a better suited example to compare to their results.

*We have now modified the text to: "As PMF analysis traditionally is used on ambient data, it is relevant to compare the findings from the AURA chamber experiments to ambient studies. Here the analysis presented by Lee et al. (2016) is relevant as they used PMF analysis to explore the SOA sources in a coniferous forest mountain region in British Columbia, where SOA concentrations reached up to 5 µg m-3 and the temperature varied from ~5 °C to ~25 °C, corresponding to the temperature in the upper range of the experiments presented in this paper. PMF factors obtained from the ambient AMS data showed a background source and two biogenic SOA sources: BSOA1 from terpenes oxidized by ozone and nitrate radical during nighttime, and BSOA2 from terpenes oxidized by ozone and OH-radical during daytime. Especially the BSOA1 O:C ratio (0.56), H:C ratio (1.56) and the overall distribution of peaks (particularly with respective to the relative ratios of m/z 58 to m/z 55 and to the ions above m/z 60) in the mass spectrum are comparable to Factor 1 (high temperature). Since both ozone and OH-radicals are present in the dark AURA chamber experiments is it interesting that the comparability to the Lee et al factor representing ozone and nitrate radical at nighttime (BSOA1) is higher than the comparability to the factor representing terpenes oxidized by ozone and OH-radical at daytime (BSOA2). It supports the previous finding, that the OH-oxidation plays a minor role in the ACCHA campaign experiments (Quéléver et al., 2019). The comparison and similarity between PMF factors from laboratory and ambient observations indicates that the PMF analysis of chamber SOA chemical composition obtained under different temperature and loading conditions can be useful for the interpretation and understanding of ambient SOA composition and vice versa."*

11) The authors should compare their results with the findings of Zhao et al. (2019)

*Thanks. We have presented the work of Zhao et al. (2019) in the introduction together with presentation of other studies what included temperature ramp experiments.*

**Specific comments:**

1) What was the time resolution of the AMS?

*The time resolution of the AMS is ~1 minute.*

2) Why was 35 min chosen to start the T ramp in Exp 1.4 and 1.5?
How much of the precursor was still left at that time?

*35 minutes was chosen to ensure that the experiments were running well at the constant temperature and comparable to experiments 1.1 and 1.3, respectively, before starting the temperature ramp.*

3) How large are the uncertainties of estimating density from O:C&H:C? (see alsoMajor comment 7). Is this uncertainty smaller than what would be expected from using average RIE&CE values? Did you collect particle time of flight (pToF) data? Could you not compare the vacuum aerodynamic diameter with SMPS data? Without an uncertainty estimate such small changes as described in Fig S9 cannot be interpreted.

*The deviation in the calculated O:C and H:C values is discussed in response to Major comment 7. The errors in mass concentration calculations (RIE and CE) do not affect the uncertainties in the O:C and H:C calculation.*

4) OSC can be calculated based on O:C and H:C. Are the trends in OSC during an experiment different from those in the density you calculate?

*The density and OSC are linearly correlated. This is now shown in the Supplementary Information.*

5) Your Q/Qexp values seem to level of at a value a lot larger than 1 which may be caused by using a too small measurement error value. Have you checked your measurement error estimate? Could there be additional sources of uncertainty?

*Yes. We did check to see if the measurement errors used in the analysis were appropriate. The measurement errors were checked against the standard deviation in time trends of ions during stable time periods and were found to agree well.*

*We have added the text below to the Supplementary Information to address additional sources of uncertainty:*

*"Additional sources of uncertainty that are not accounted for in the PMF analysis of high-resolution mass spectra are uncertainties related to HR fitting, including errors in peak shape, and m/z calibrations (Cubison et al. (2015))."*

6) In addition to Q/Qexp values the ratio of explained variance to total variance may be worth looking at when discussing which PMF solution was selected.

*Thank you for this suggestion. We have looked into this, but find that the dependence of the Q/Qexp vs. time and m/z plots as factors are added is more useful. Thus we have included these graphs as a part of the response to question 9.*

7) The mass spectra presented in e.g. Fig 1b or Fig 3 look like the HR data was recombined to UMR resolution and colour coded by family. If this is the case, this needs to be clearly stated.

*HR data is used for the mass spectra.*

8) Did you really use m/z 43 and m/z 44 derived from UMR analysis (using UMR fragtables) for the tracer ion analysis (page 8)? Or did you run the HR analysis and recombine to UMR signals? This needs to be stated.

*In section 3.2.2.it is stated:*
*"Figures 6a and 6b are "triangle plots" (Ng et al. 2010) showing f44 (the fraction of m/z 44 relative to the total mass in the spectra) as a function of f43 (the fraction of m/z 43 relative to the total mass in the spectra) obtained from unit mass resolution data from the AMS".*

9) For the second set of ramp experiments (Exp 1.5 and 1.6), it is of interest how much of the gas phase chemistry had already happened when you started the ramps as there should be some temperature dependence for the kinetics. Was all a-pinene consumed? Also, what were the starting gas phase compositions? Was the same amount/type of HOMs formed?

*We assume the reviewer refers to experiments 1.4 and 1.5. In the 10 ppb α-pinene experiments the gas phase concentrations were too low to do CIMS measurements or to monitor the VOC using GC-FID. Therefore we cannot conclude on the HOMs composition in these experiments.*

10) As the numbering of Factors in the PMF solutions does not have a meaning (to my knowledge), you should consider a meaningful numbering, e.g. by the (assumed) volatility of the factors or the degree of oxidation (OSc). This will make the factor description and comparison easier to understand for the reader.

*We have now changed the order of the factors according to the comment to reflect trends in estimated C\**

11) It is difficult to see how the relative contribution of the factors change with time in Fig 2. Add the "unstacked" factor time series to the SI material. This will help especially with interpreting the T ramp experiments.

*The time series of the unstacked factors from each panel in Figure 2 are added to the supplementary information.*

12) What happened in Exp 3.3 around 50 min? There is a bump/shoulder in the timeseries.

*It is not clear what happened, but we do not expect that it affects the outcome of the analysis and the resulting conclusions. We have added the following sentence to the manuscript: "the reason for the bumb/shoulder at around 50 min in Exp. 3.3. is unclear."*

13) When you are interpreting the simultaneous O:C and H:C change for the 50ppb experiments (page 7 last paragraph) you should take into account that in this specific case there is very little change in OSC, i.e., that the average degree of oxidation does not change. This may point towards non-oxidative reactions being the main source for the composition change. Or it might mean that at the same time more oxidised compounds (higher O:C than already in particles) are condensing as they are formed by ongoing oxidation reactions in the gas phase and also more volatile compounds (higher H:C than already in the particles) partition due to the increasing cOA. The netchange could then be as you observe.

*We have now modified the section about O:C and H:C and added the OSC.*

14) If you argue that compounds condensing at low temperatures are not well captured by the elemental analysis parameterisation, that would also be relevant for ambient measurements. 0∘C is not that uncommon during wintertime or nights, and the SOA mass loadings in your high concentration experiments are not that far off from ambient loadings in more polluted regions. Is it not more likely that you are simply observing condensation of more volatile and less oxidized species in these experiments? The 50 ppb and -15∘C case should have the least volatile compounds condensing as cOA was highest and C* values will be lowest. At higher temperatures, the effect vanishes as that compound class no longer partitions into the particle phase. You may just be probing a different range compounds at the lowest temperature (i.e., compounds that are too volatile to condense at 20∘C).

*Upon further consideration, we have removed the argument related to elemental analysis parameterization. While composition differences between species that condense at the lower temperatures and those that were used to parameterize the O:C and H:C may be significant enough to affect the accuracy of those values, this does not serve to explain the trend in observed H:C and O:C at the low temperature.*

15) Why were the T ramps during SOA formation done at 10 ppb and the T ramps at the end of the experiments at 50 ppb?

*As explained above the ramps at the end of the 50 ppb experiments are omitted from the revised manuscript.*

16) The slope of the O:C values for Exp 2.2 and 2.3 seems to change when T crosses 0∘C. For Exp 2.1 it is hard to say. Could there be an effect of "frozen" particles?

*As mentioned above we have omitted the 50 ppb ramps from the manuscript.*

17) Figs 4b and 6b reveal some differences between "same" experiments which are not addressed in the text. Exp 2.1 and 3.1 show a bigger difference between each other than Exp 2.1 and 1.1 (Fig 5). In Fig 6, Exp 2.1 data is not visible. However, Exp 2.2 has decreasing f44 and f43, but Exp 3.2 is constant. f44 decreases in Exp 3.3 as much as it increases in Exp 2.2. I would not call that "good reproducibility" (page 8).

*We would like to point out that the Figure 6b is an expansion of Figure 6a. Based on the comment from the reviewer we have modified the paragraph and been more cautious in the formulations.*

18) In Fig 6b, the arrow indicates a decrease of f43 for Exp 3.2 but the text states an increase for all experiments at -15∘C. Or is the arrow the wrong way round?

*The direction of the arrow related to Exp. 3.2, which is conducted at 0 °C, is correct.*

19) How much of the SOA mass is explained by the acids and esters detected in LC-MS? It is not simply (18+4)% and (38+11)%

*The mass fractions of acids and esters at the different temperatures and different VOC loadings are explicitly stated in Kristensen et al. 2020.*

**Language and presentation:**
+Decide if you want to separate adverbial terms at the start of a sentence by a comma. In many cases, you used a comma. In other cases, you did not.

*We have made the use of commas more consistent.*

+Reference entries to Quelever and Rosati are missing.

*Thank you. They are now added.*

+page 2 line 8 "and has it been" -> "and it has been"

*It is now corrected.*

+ page 2 line 10 OH and NO3 are not negative ions but radical.  Use "·" (dot) not "-"

*Thanks. Corrected.*

+ page 2 line 17f put the e.g part into commas or remove comma before "because"

*The e.g. part is now put into commas.*

+ page 4 line 35 "The number of factors are chosen..." -> is chosen

*Thanks. Corrected.*

+ page 5 line 21&23, page 6 line 14 you use the term "CHO" in contrast to CHOgt1.But later you use "CHO1" several times. -> Choose one label.

*It is now changed to CHO1 at all places.*

+ page 6 lines 21-28 this should be together with the previous paragraph as it is still describing the factor MS. Do not just start a new paragraph become the current one is feeling too long.

*We have modified the order of the content  which means that this comment is no longer relevant.*

+ page 7 line 36/37 "hydration reactions with carbonyls" -> hydration reactions of car-bonyls

*Thanks. This is now corrected.*

+ page 8 line 30 "Alfarra 2004" -> Alfarra et al. 2004

*In the reference list we are having both Alfarra 2004 and  Alfarra et al. (2004). In this specific case we are referring to Alfarra (2004).*

+ page 8 line 40 "No clear tendencies" -> should be trends in this context

*Corrected.*

+ page 9 line 5 "(experiments 1.2, 2.2, and 2.3)" -> should be 3.2?

*Yes. Corrected.*

+ page 9 line18f "...to identify and quantify the 10 carboxylic acids which are regarded as some of the most important in α-pinene derived SOA..." This sentence does notwork. Most important what? Constituents?

*This is now changed to: "As described in detail in Kristensen et al. (2019) LC-MS analysis was performed to identify and quantify 10 carboxylic acids formed in the darks ozonolysis of α-pinene and in current experiments constituting 18-38 % of the SOA mass…"*

+ Fig 4: Are the legend descriptions of Exp3.2&3.3 switched?

*Yes. Corrected.*

+ Fig 4: The black stroke around dark blue triangles is very hard to see -> use lighter color for -15∘C experiment (same issue in Fig 6).

*We have slightly modified the color scale to make it more visible.*

+ Fig 4b: Do not put the yellow arrow on top of blue triangles

*The yellow arrow is moved so do not cover the blue triangles.*

+ Fig 5: You use Kelvin as temperature unit while the text and labels use∘C. -> Choose one unit.

*The temperature is now changed to °C.*

+ Fig 5: The yellow dashed line is hidden by blue diamonds -> adjust y scale a bit to make it visible.

*The figure is now adjusted to make the content more visible.*

+ Fig 6b: where are the Exp3.1 data points? I only see red triangles with black border.

*The Exp. 3.1 data points are now visible.*

+ Fig 7 The term AMS m/z 44 *[factor] is a little bit confusing as the text speaks of(PMF) factors. maybe change this to [correction factor] or [scaling factor]+ all SI Fig with PMF results: y axis is mass concentration not Mass. That is "lab-slang". You can easily avoid too long y axis label by introducing an abbreviation for mass concentration (e.g. cm or corg).

*To prevent confusion we have now changed the use of "[factor]" to "[scaling factor]". "Mass" on the y-axis is changed to "mass concentration".*

+ all SI Fig with PMF results: experiment labels are 1-8 which is different from maintext -> adjust labels.

*The labels in the SI are now the same as in the main text.*

References:

Bougiatioti, A., Stavroulas, I., Kostenidou, E., Zarmpas, P., Theodosi, C., Kouvarakis,G., Canonaco, F., Prévôt, A. S. H., Nenes, A., Pandis, S. N. and Mihalopoulos, N.: Pro-cessing of biomass-burning aerosol in the eastern Mediterranean during summertime,Atmos. Chem. Phys., 14(9), 4793–4807, doi:10.5194/acp-14-4793-2014, 2014.

Bruns,E. A., El Haddad, I., Keller, A., Klein, F., Kumar, N. K., Pieber, S. M., Corbin, J. C.,Slowik, J. G., Brune, W. H., Baltensperger, U. and Prévôt, A. S. H.: Inter-comparisonof laboratory smog chamber and flow reactor systems on organic aerosol yield andcomposition, Atmos. Meas. Tech., 8(6), 2315–2332, doi:10.5194/amt-8-2315-2015,2015.

Cubison, M.J. and Jimenez, J.L., 2015. Statistical precision of the intensities retrieved from constrained fitting of overlapping peaks in high-resolution mass spectra. *Atmospheric Measurement Techniques (Online)*, *8* (6).

Kostenidou, E., Lee, B.-H., Engelhart, G. J., Pierce, J. R. and Pandis, S. N.:Mass Spectra Deconvolution of Low, Medium, and High Volatility Biogenic SecondaryOrganic Aerosol, Environ. Sci. Technol., 43(13), 4884–4889, doi:10.1021/es803676g,2009.

Kristensen, K., Jensen, L. N., Glasius, M. and Bilde, M.: The effect of sub-zerotemperature on the formation and composition of secondary organic aerosol fromozonolysis of alpha-pinene, Environ. Sci. Process. Impacts, 19(10), 1220–1234,doi:10.1039/c7em00231a, 2017.

Kristensen, K., Jensen, L., Quéléver, L. L., Chris-tiansen, S., Rosati, B., Elm, J., Teiwes, R., Pedersen, H., Glasius, M., Ehn, M.and Bilde, M.: The Aarhus Chamber Campaign on Highly Oxidized MultifunctionalOrganic Molecules and Aerosols (ACCHA): Particle Formation and Detailed Chem-ical Composition at Different Temperatures, Atmos. Chem. Phys. Discuss., 1–22,doi:10.5194/acp-2020-99, 2020.

Kroll, J. H., Donahue, N. M., Jimenez, J. L., Kessler, S. H., Canagaratna, M. R., Wil-son, K. R., Altieri, K. E., Mazzoleni, L. R., Wozniak, A. S., Bluhm, H., Mysak, E. R., Smith, J. D., Kolb, C. E. and Worsnop, D. R.: Carbon oxidation state as a metric fordescribing the chemistry of atmospheric organic aerosol, Nat. Chem., 3(2), 133–139,doi:10.1038/nchem.948, 2011.

Lee, A. K. Y., Abbatt, J. P. D., Leaitch, W. R., Li, S. M., Sjostedt, S. J., Wentzell, J. J. B.,Liggio, J. and Macdonald, A. M.: Substantial secondary organic aerosol formation in aconiferous forest: Observations of both day- and nighttime chemistry, Atmos. Chem.Phys., 16(11), 6721–6733, doi:10.5194/acp-16-6721-2016, 2016.

Li, Z., Tikkanen, O.P., Buchholz, A., Hao, L., Kari, E., Yli-Juuti, T. and Virtanen, A.: Effect of DecreasedTemperature on the Evaporation ofα-Pinene Secondary Organic Aerosol Particles, ACS Earth Sp. Chem., 3(12), 2775–2785, doi:10.1021/acsearthspacechem.9b00240,2019.

Quelever, L. L. J., Kristensen, K., Normann Jensen, L., Rosati, B., Teiwes, R., Dael-lenbach, K. R., Peräkylä, O., Roldin, P., Bossi, R., Pedersen, H. B., Glasius, M., Bilde,M. and Ehn, M.: Effect of temperature on the formation of highly oxygenated organicmolecules (HOMs) from alpha-pinene ozonolysis, Atmos. Chem. Phys., 19(11), 7609–7625, doi:10.5194/acp-19-7609-2019, 2019.

Wan, K. X., Vidavsky, I. and Gross, M. L.: Comparing similar spectra: From simi-larity index to spectral contrast angle, J. Am. Soc. Mass Spectrom., 13(1), 85–88,doi:10.1016/S1044-0305(01)00327-0, 2002.

Zhao, Z., Le, C., Xu, Q., Peng, W., Jiang, H., Lin, Y.-H., Cocker, D. R. and Zhang, H.: Compositional Evolution of Secondary Organic Aerosol as Temperature and RelativeHumidity Cycle in Atmospherically Relevant Ranges, ACS Earth Sp. Chem., 3(11),2549–2558, doi:10.1021/acsearthspacechem.9b00232, 2019